# Evolutionary dynamics of genome size and content during the adaptive radiation of Heliconiini butterflies

Francesco Cicconardi [1,2] ✉, Edoardo Milanetti[3,4], Erika C. Pinheiro de Castro [2], Anyi Mazo-Vargas [5], Steven M. Van Belleghem [6,7], Angelo Alberto Ruggieri [6], Pasi Rastas [8], Joseph Hanly [9,10], Elizabeth Evans[6], Chris D. Jiggins[2], W. Owen McMillan[10], Riccardo Papa [6,11,12], Daniele Di Marino[13,14,15], Arnaud Martin [9] & Stephen H. Montgomery [1,10] ✉

*Heliconius* butterflies, a speciose genus of Müllerian mimics, represent a classic example of an adaptive radiation that includes a range of derived dietary, life history, physiological and neural traits. However, key lineages within the genus, and across the broader Heliconiini tribe, lack genomic resources, limiting our understanding of how adaptive and neutral processes shaped genome evolution during their radiation. Here, we generate highly contiguous genome assemblies for nine Heliconiini, 29 additional reference-assembled genomes, and improve 10 existing assemblies. Altogether, we provide a dataset of annotated genomes for a total of 63 species, including 58 species within the Heliconiini tribe. We use this extensive dataset to generate a robust and dated heliconiine phylogeny, describe major patterns of introgression, explore the evolution of genome architecture, and the genomic basis of key innovations in this enigmatic group, including an assessment of the evolution of putative regulatory regions at the *Heliconius* stem. Our work illustrates how the increased resolution provided by such dense genomic sampling improves our power to generate and test gene-phenotype hypotheses, and precisely characterize how genomes evolve.

A central goal of evolutionary biology is to understand how biodiversity is generated, maintained, and how interactions between organisms drive the diversity of natural communities. Periods of rapid diversification are often associated with the colonization of new ecological niches or the exploitation of new resources[1]. The evolution of key innovations, such as physiological adaptation to food resources, or new morphological traits, can enable these ecological shifts, and play critical roles in adaptive radiations[2]. From a genetic perspective,

[1]School of Biological Sciences, Bristol University, Bristol, United Kingdom. [2]Department of Zoology, University of Cambridge, Cambridge, United Kingdom. [3]Department of Physics, Sapienza University, Piazzale Aldo Moro 5, 00185 Rome, Italy. [4]Center for Life Nano- & Neuro-Science, Italian Institute of Technology, Viale Regina Elena 291, 00161 Rome, Italy. [5]Department of Ecology and Evolutionary Biology, Cornell University, Ithaca, NY 14853, USA. [6]Department of Biology, University of Puerto Rico, Rio Piedras, PR, Puerto Rico. [7]Ecology, Evolution and Conservation Biology, Biology Department, KU Leuven, Leuven, Belgium. [8]Institute of Biotechnology, University of Helsinki, Helsinki, Finland. [9]Department of Biological Sciences, The George Washington University, Washington DC, WA 20052, USA. [10]Smithsonian Tropical Research Institute, Panama City, Panama. [11]Molecular Sciences and Research Center, University of Puerto Rico, San Juan, PR, Puerto Rico. [12]Comprehensive Cancer Center, University of Puerto Rico, San Juan, PR, Puerto Rico. [13]Department of Life and Environmental Sciences, New York-Marche Structural Biology Center (NY-MaSBiC), Polytechnic University of Marche, Via Brecce Bianche, 60131 Ancona, Italy. [14]Neuronal Death and Neuroprotection Unit, Department of Neuroscience, Mario Negri Institute for Pharmacological Research-IRCCS, Via Mario Negri 2, 20156 Milano, Italy. [15]National Biodiversity Future Center (NBFC), Palermo, Italy. ✉e-mail: francicco@gmail.com; s.montgomery@bristol.ac.uk

one fundamental question in understanding how adaptive radiations emerge, is if a significant amount of change, and sources of variability, originate prior to the acceleration in diversification, and whether this variation facilitates the subsequent adaptive radiation. This would be consistent with phyletic gradualism at a genetic level. Identifying and understanding the genetic basis of such key innovations is now a realistic goal[3,4], and can provide explicit links between genetic changes, natural selection and speciation, in the context of wider patterns of genomic divergence.

Heliconiini, a Neotropical tribe of Nymphalid butterflies, comprised of ~80 species and ~400 subspecies, have become a key system to explore the biology of speciation[5–8]. In particular, the rapid radiation of the genus *Heliconius*, and their diversity of color patterns, have become a case study in how genomic approaches can improve our understanding of the genetic architecture of adaptive traits, and the accumulation of reproductive isolation with ongoing gene flow[7]. Heliconiini also exhibit key innovations including the tribe-wide restricted use of Passifloraceae as larval hostplants – their antagonist coevolutionary partners that can provide them with cyanogenic glucosides for chemical protection[9]. Within the Heliconiini, species of the genus *Heliconius* are also the only lepidoptera to actively collect and digest pollen as adults, which is associated with major shifts in reproductive lifespan[10], and behavioral and neural elaboration[11]. As such, the availability of tribe-wide genomic resources would represent a major resource to explore the biology of an enigmatic case study in adaptive diversification.

Here, we provide such a resource by sequencing and assembling genomic data, and using a combinatorial approach to maximize methodological outcomes, with a unified cross-species annotation to remove possible species-biases previously unrepresented in available data. Combined with already available resources, which we also improve both in terms of assembly contiguity and gene annotation, we generate a genomic dataset that comprises ~75% of all the species in the Heliconiini tribe, to our knowledge one of the most comprehensive efforts to sample an insect tribe at high taxonomic density. With this, we produce a comprehensive dated phylogeny for Heliconiini and explore patterns of gene flow across the tribe. We test if a significant and substantial amount of genomic change occurred not only at the stem of *Heliconius*, but also at more basal branches within the Heliconiini tribe, pre-dating the range of innovations seen in *Heliconius*. Finally, we investigate structural and adaptive aspects of genome evolution across the radiation and during key ecological transitions, and explore evidence of accelerated evolution in putative regulatory elements. Our analyses provide refined views of genomic diversity across Heliconiini, and provide new gene-phenotype hypotheses that will provide the foundation for future functional experiments.

## Results & discussion

### Improved resolution of phylogenetic relationships and signatures of introgression across the genome

To generate the species tree, we first compiled a total data set of 4,011,390 base pairs of aligned protein-coding DNA obtained from the single-copy orthologous groups (scOGs) obtained from a high quality and revised genomic resources (Supplementary Note 1, Supplementary Figs. 1–19). The alignment has over 1.5 M parsimony-informative, ~500k singleton sites, and 1.9k constant sites. A species-level phylogeny was determined with a maximum-likelihood (ML) analysis (Supplementary Figs. 19 and 20), and used to estimate divergence dates (Fig. 1c. Supplementary Fig. 21, and Supplementary Data 3). Although the topology is widely consistent with previously inferred phylogenetic relationships[8], we identify differences within some *Heliconius* clades, where the Silvaniform and Melpomene clades are now paraphyletic, and among other genera of Heliconiini, with *Podothricha telesiphe* and *Dryas iulia* now sister lineages, outgrouped by *Dryadula phaetusa* and *Philaethria dido*. The estimated divergence times show that the subfamily

Heliconiinae originated ~45.3 million years (Mya) (95% CI: 35.9–55.5), with the last common ancestors of *Eueides* and *Heliconius* dating to ~11.1 Mya (95% CI: 7.3–12.1) and 9.6 Mya (95% CI: 8.8–13.8), respectively. Interestingly, deeper branches of the phylogeny are characterized by high molecular substitution rates (Fig. 1c and Supplementary Data 3), indicating a series of bursts in evolutionary rate at the base of the radiation, supported by a highly sampled posterior distribution across our tree (ESS ≫ 1000; Supplementary Fig. 21). To account for incomplete lineage sorting (ILS) within the phylogeny, we used a coalescent summary method for species trees reconciliation using gene trees (Fig. 2b). This resulted in an almost identical topology as the ML tree (Fig. 1c, Supplementary Figs. 19 and 20), with a single exception of the *H. clysonymus* + *H. hortense* + *H. telesiphe* branch, which could be due to high rates of ILS or introgression (coalescent units = 0.08), disrupting the monophyly of the Erato clade[12]. We find little evidence of ILS around more basal nodes, with the percentage of quartets in gene trees that agree with the ML topology (normalized quartet support) q1 (f1) being 0.62 (1989); higher than nodes supporting other deep splits in *Heliconius* (Doris + Wallacei + Silvaniform + Melpomene clade, with the Wallacei + Silvaniform + Melpomene branch) (Supplementary Note 2).

*Heliconius* have also become key taxa for exploring the impact of gene flow and hybridization on adaptive divergence[5,7,13]. We therefore revisited this topic with our extended taxonomical range, adopting two very recent methodological approaches: the discordant-count test (DCT) and the branch-length test (BLT). Both tests reveal a lack of gene flow between basal Heliconiini nodes and those at the base of the *Heliconius* radiation, including the *H. aoede* split, but do identify several introgression events within major clades of Heliconiini (Fig. 2, and Supplementary Data 4, 5). Note, the putative lack of introgression at the basal node of Heliconiini is unlikely to be simply explained by a lack power in the statistical methods used to detect introgression. The Heliconiini split is dated between 20 and 30 Mya, and the same methodology, applied to the *Drosophila* radiation[14], has identified introgression events dated over 20 My, suggesting that in principle the methods applied should be able to find introgression in our phylogenetic framework. The greatest number of introgression events were detected within *Heliconius*, specifically between the most recent common ancestors (MRCAs) of Erato + Sara/Sapho clade, the Doris + Wallacei + Silvaniform + Melpomene clade, and within the Erato clade. Interestingly, the Sara/Sapho clade shows very low rates of introgression, potentially reflecting a stronger barrier to gene flow[15] between species in this clade, where females mate only once (monoandry), and males often mate with females as they eclose from the pupae (referred to as pupal mating)[16]. Across all branches, the estimated fraction of introgressed genome mostly varies between 0.02 γ to 0.15 γ, with a peak around 0.30 within the Erato clade (range of average γ estimates = 0.023–0.323). Most introgression events also occurred in a restricted time frame within the last 5 Mya (Fig. 2b), and no significant relationship was found between the midpoint estimate of the timing of introgression and the estimated γ (Fig. 2b), indicating that the fraction of a genome that is introgressed within *Heliconius* does not depend on the timing of those introgression events (see Supplementary Note 2 for more details).

### The origin of major *Heliconius* lineages and pollen-feeding

Pollen-feeding is one of the most important key innovations within *Heliconius* radiation. So far, all phylogenetic reconstructions based on molecular data[5,8] place the non-pollen feeding clade Aoede (members of the genus formerly known as *Neruda*) within the *Heliconius* clade, suggesting a secondary loss in this lineage. The comparison of this lineage, represented in our data by *H. aoede*, with the pollen-feeding *Heliconius* species offers the possibility to understand the genetic basis of the traits related to pollen-feeding and potentially to solve the puzzle about its emergence. Specifically, we can test whether *i)* pollen-feeding emerged once, at the stem of *Heliconius*, with the Aoede clade

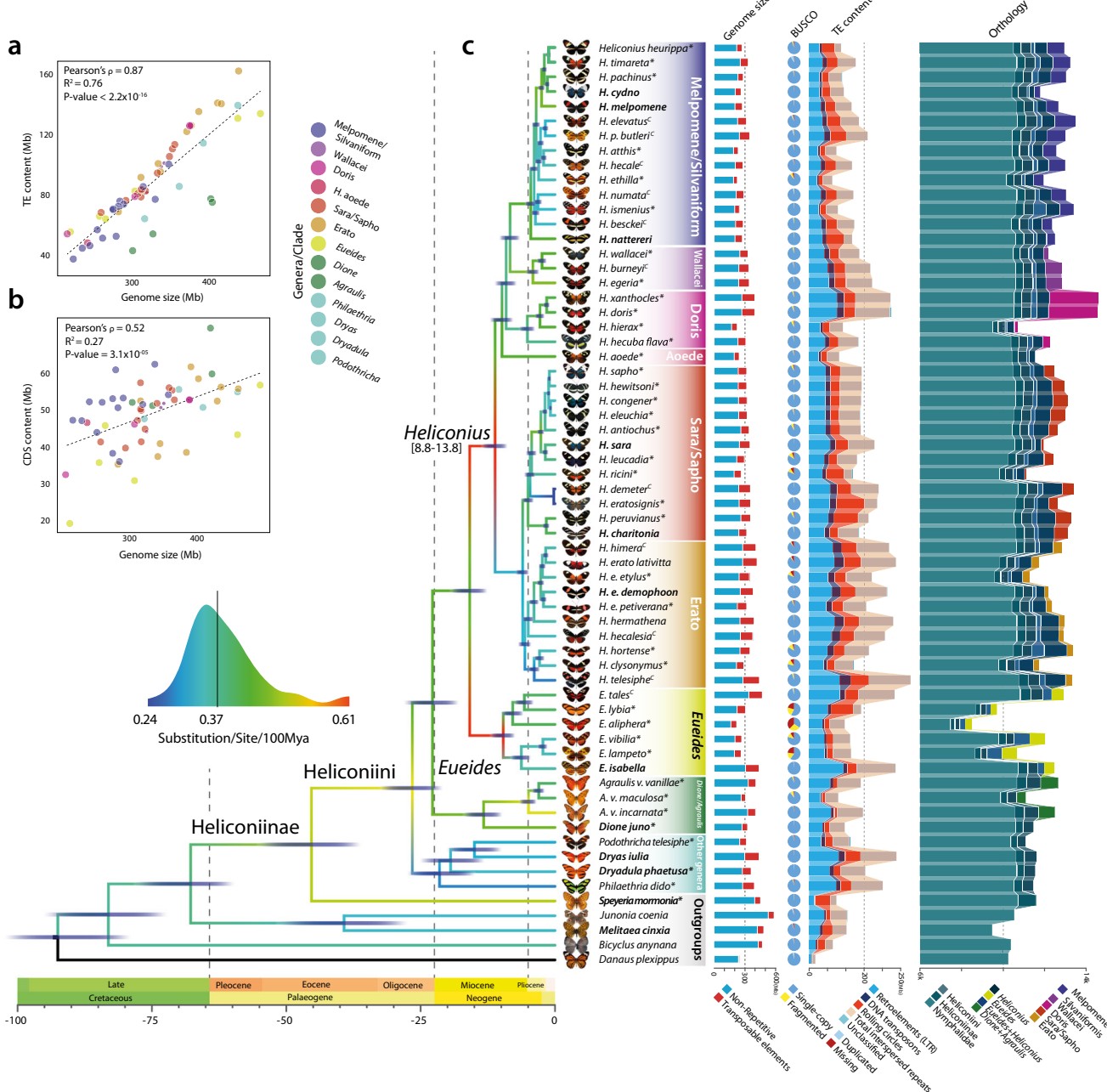

**Fig. 1 | Phylogenetic, genomic, and proteomic comparisons among 63 Nymphalid butterfly species. a, b** The contribution of transposable elements (TEs) and coding regions (CDS) to genome size variation across Heliconiinae, respectively. Data were fitted to a linear model (lm; *P* values ≤ 2.2 × 10⁻¹⁶). **c** From left to right: i) the dated species phylogeny built from the concatenated single-copy orthologous groups (scOGs) from all sequenced Heliconiinae and outgroups, using a combination of Maximum Likelihood and Bayesian Inference. The branch color represents the number of substitutions per site per 100 Mya of that specific branch. Species names in bold indicate the species with chromosome- or sub-chromosome-level assemblies, asterisks indicate genomes assembled in this study, ᶜ curated assemblies; ii) genome assembly size, in red the TE fractions; iii) BUSCO profiles for each species. Blue indicates the fraction of complete single-copy genes; iv) bar plots show total gene counts partitioned according to their orthology profiles, from Nymphalids to lineage-restricted and clade-specific genes.

outside *Heliconius s.s.*; *ii*) or if it emerged once with Aoede falling within *Heliconius*, and was secondarily lost in the Aoede clade; or *iii*) if it evolved independently in the Erato and Melpomene clades, with Aoede falling within *Heliconius* but without invoking trait loss. Using extensive genomic data in the form of scOGs, our data support the monophyletic status of the pollen-feeding *Heliconius* + *H. aoede* (Fig. 1c, and Supplementary Fig. 19). Specifically, *H. aoede* seems to cluster sister to the stem of three other clades: Melpomene/Silvaniform, Wallacei and Doris (Figs. 1c and 2b). This position is strongly supported by bootstrap and concordance values (Fig. 1c and

Supplementary Figs. 19 and 20). A further assessment of nodal support was performed using the Quartet Concordance (QC), Quartet Differential (QD) scores, and Quartet Informativeness (QI) (within Quartet Sampling) to identify quartet-tree/species-tree discordance (see Methods). The position of *H. aoede* remained supported, with a strong majority of quartets supporting the focal branch (QC = 0.9), with a low skew in discordant frequencies (QD = 0) indicating that no alternative history is favored, no signal of introgression is detected (i.e., QD <1 but >0), and a QI of 1 indicates that the quartets passed the likelihood cut-off in 100% of the cases (Supplementary Figs. 22 and 23; QC = 0.9).

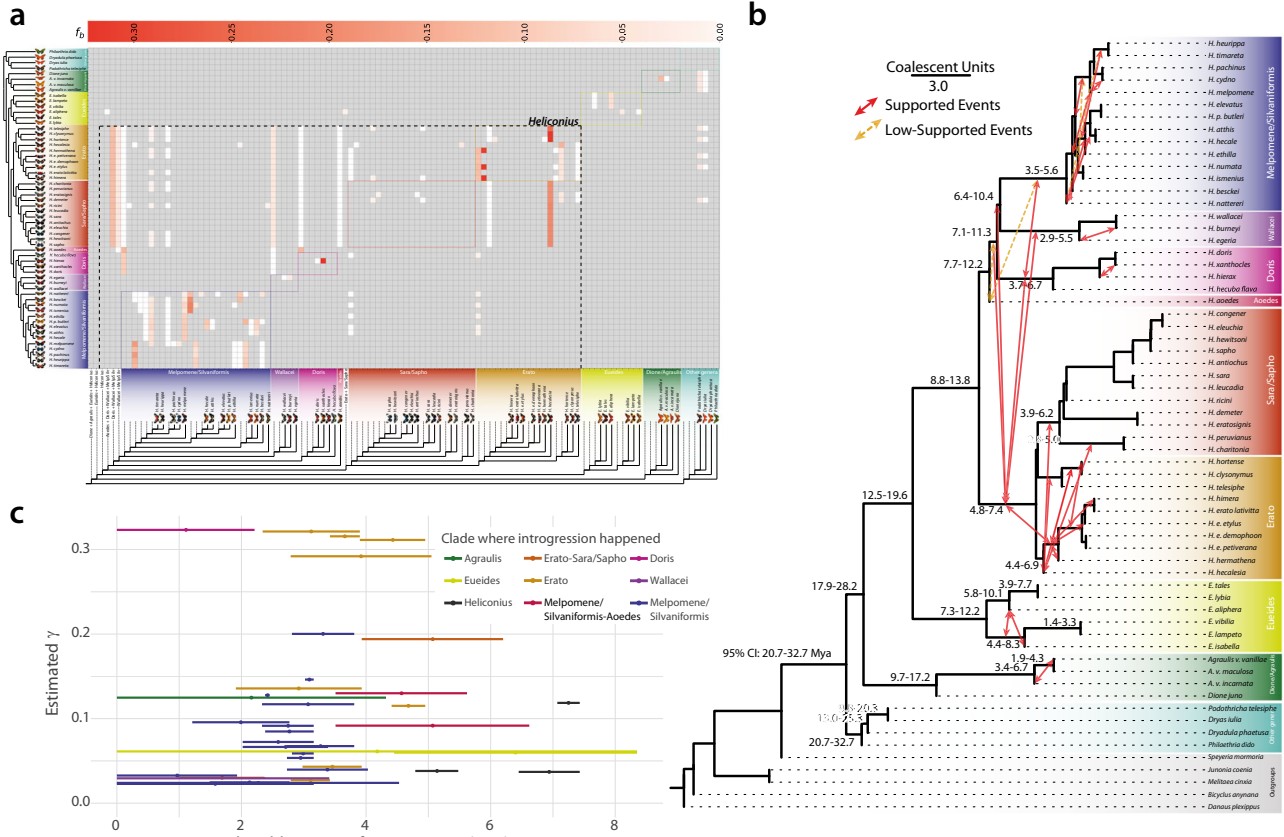

**Fig. 2 | Patterns of introgression inferred for the Heliconiinae clade. a** The matrix shows inferred introgression proportions as estimated from scOG gene trees in the introgressed species pairs, and then mapped to internal branches using the f-branch method. The expanded tree at the bottom of the matrix shows both the terminal and ancestral branches. **b** ASTRAL-III species tree derived from nucleotide gene trees, with mapped introgression events (red arrows) derived from the corresponding f-branch matrix. Yellow dashed arrows indicate introgression events with lower support (triplet support ratio <10%). Branch lengths correspond to coalescent units. Numbers on nodes correspond to the confidence interval of the dated phylogeny (Fig. 1e). Note how, not only, most of the introgression events happen within clades and among time overlapped nodes, but also how the majority of introgressive events are affecting lineages with low CUs, indicating a lower barrier to gene flow. There seems to be only one introgression for *H. aoede*, which happened with the Silvaniform/Melpomene basal branch. Times inferred from the dating analysis summarized in Fig. 1. **c** Each segment indicates the confidence interval (CI 95% from Fig. 1c) of the estimated introgression event (triplet support ratio >10%). The circles indicate the average date.

Leveraging the 63-way whole genome alignment and using *E. isabella* as reference, we further tested the robustness of this topology by inferring the local topology history across the 63 species. We generate non-overlapping windows of 10 kb across the whole 63-way whole genome alignment and use them to infer ML trees at each window, exploring the frequency of possible topologies, the effect of introgression and incomplete lineage sorting (ILS) with a coalescent based method. From more than 43k non-overlapping sliding windows, ~30k returned one of five main topologies. With the only purpose of exploring the monophyly of the pollen-feeding trait, we classified the topologies based on the position of *H. aoede* relative to the other *Heliconius* clades, *Eueides* and other non-*Heliconius* species (Supplementary Fig. 24a). The most frequent/supported topology (Topology 1, 49% of trees), shows the same relationships of our species tree reconstruction. Less frequent topologies (Topology 2,4, and 5, total of 16% of trees) also show *H. aoede* nested within *Heliconius*, while Topology 3, the topology that places *H. aoede* outside *Heliconius s.s.*, has a frequency of 3.7% (see Supplementary Note 3 and Supplementary Fig. 24). Aware of the possible impact of introgression and ILS on topology inference, we used the same non-overlapping sliding windows to infer the impact of those on different chromosomes, expecting Z chromosome to be less affected by both[7]. The fraction of the genome that introgressed (average *f*-branch statistic) across all triplet comparisons and coalescent units (CUs), as a proxy of ILS, from

each chromosome and the Z chromosome versus all autosomes (see Supplementary Note 3 and Supplementary Figs. 25, 26) show that, indeed, the Z chromosome has a lesser degree of introgression and ILS overall, but this effect does not change the topologies' frequencies in favor of Topology 3, which stays ~5% of the entire chromosome, whereas Topology 1 increase to 56%.

Overall, given the methods currently available for large phylogenomic datasets such as ours (but see Thawornwattana et al.[17]), the landscape of local history seems to confirm the species tree as the most consistent topology, with *H. aoede* clustering within *Heliconius* clades. This would likely exclude a single gain of pollen-feeding with no loss (*i*), leaving two nominally equally parsimonious scenarios; one gain at the stem of *Heliconius* clade, followed by one loss at the branch of *H. aoede*, or two independent gains at the base of Sara-Sapho/Erato and Doris/Wallacei/Melpomene/Silvaniform. For our purposes, this provides a hypothesis testing framework where, under the first scenario which is traditionally seen as most likely[10], signatures of molecular innovation relating to the suite of traits linked to pollen-feeding may be expected to occur on the stem *Heliconius* branches, while the pattern of evolution on *H. aoede* is predicted to be linked to trait loss. In what follows, we use our phylogenetic framework to explore the evolution of genomic size, content and patterns of selection in key points of the Heliconiini radiation.

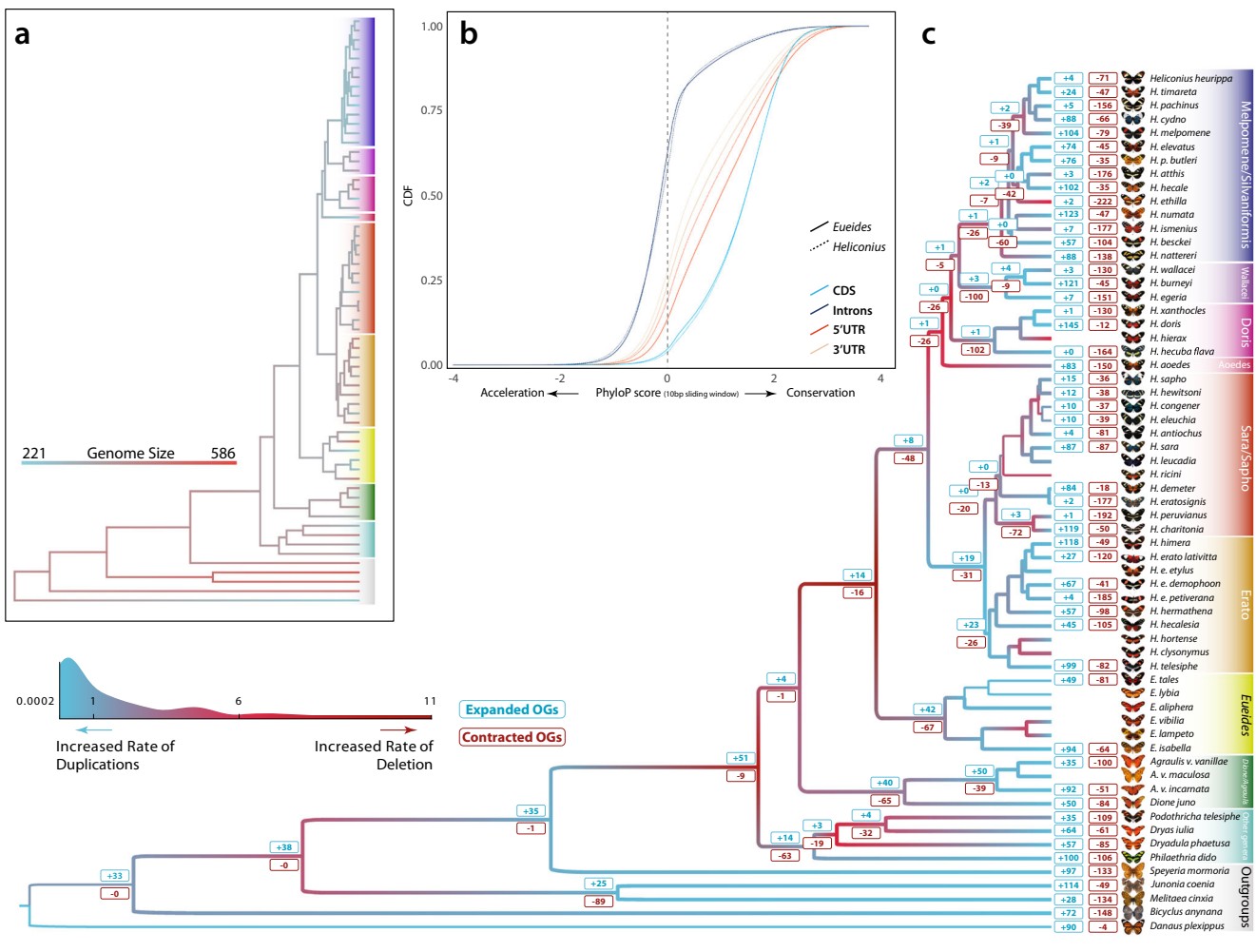

**Fig. 3 | Genomic dynamics, acceleration/conservation rates and ortholog copy number evolution. a** Ancestral genome size reconstruction of Nymphalids inferred by ML approach. **b** Cumulative distribution functions (CDFs) of scores for selected annotation classes (CDS, introns, 5' and 3' UTRs) as computed by the subtree scores for *Eueides* and *Heliconius* clades. PhyloP scores at sites of different annotation classes, based on the LRT method and multiple whole 63-species genome alignment. Positive scores indicate conservation, and negative scores indicate acceleration (CONACC mode) in a 10-bp sliding window. **c** Branch colors indicate the ratio between the rate of duplications (duplicated Mb/Mya per branch) and deletions (deleted Mb/Mya per branch) across the Nymphalid phylogeny. In general, red shifts indicate an increased rate of deletion over rate of duplication; the opposite is true for blue shifts. Numbers at nodes correspond to the amount of expanded (blue) and contracted (red) ortholog groups (values are shown for main branches and most complete genomes, see Methods).

## Evolution of genome size and content

Variation in genome size can be formalized in an "accordion" model[18] where genomes gain, lose, or maintain its size in equilibrium in each species, due to a balance between expansions in transposable elements (TEs) and large segmental deletions. By reconstructing ancestral genome sizes at each node in the Heliconiini phylogeny, we found that the MRCA of Heliconiinae experienced a 30% contraction from ~406 Mb at stem of Heliconiinae to ~282 Mb for Melpomene/Silvaniform clade; while, at the same time, other branches leading to *Philaethria*, *Dryadula*, *Dryas* and *Podothricha*, and the Erato, Doris and Wallacei clades within the genus *Heliconius*, had independent expansions. Strikingly, *H. aoede* shows a loss of about 68 Mb, a fifth of its genome size (-22%) from its ancestral node (Fig. 3, Supplementary Figs. 27–32).

There is a remarkable difference is species richness between the two sister genera, *Heliconius* and *Eueides*, but both seemed to have experienced an accelerated rate of substitution (Fig. 1c). We explicitly tested which genomic compartments (CDS, introns, 5'-UTR, and 3'-UTR) contribute to the change in substitution rate. We did so by calculating CONACC scores and assessing departures from neutrality

(Fig. 3b). Between the two genera, we identified an enrichment for higher CONACC scores in *Heliconius* for CDS and introns, compared to the same compartments in *Eueides*, a trend that is inverted for the two UTR regions (Wilcoxon rank-sum test 'two-sides' *P* value < 2.2 × 10⁻¹⁶). This suggests an increased tendency for clade-specific selection, also confirmed by the fast-unconstrained Bayesian approximation method (FUBAR), which showed that *Heliconius* have more sites under purifying selection and positive selection per codon compared with *Eueides* (Supplementary Fig. 33). *Heliconius* has 2.5x more sites under purifying selection per codon than *Eueides*, suggesting that the higher CONACC scores in *Heliconius* are likely due to higher degrees of purifying selection.

We next explored the relationship between transposable elements (TEs) and genome size, and their effect on gene architecture. We found that larger genomes tend to have a distribution of intron length skewed towards longer introns (Supplementary Fig. 18a), with a positive correlation between median intron length and total TE content (Supplementary Fig. 34; Pearson's ρ = 0.72; R² = 0.51). Long introns also accumulate significantly more TEs then expected by their size (Supplementary Fig. 35; Wilcoxon rank-sum test *P*-value = 2.13 × 10⁻¹³), with

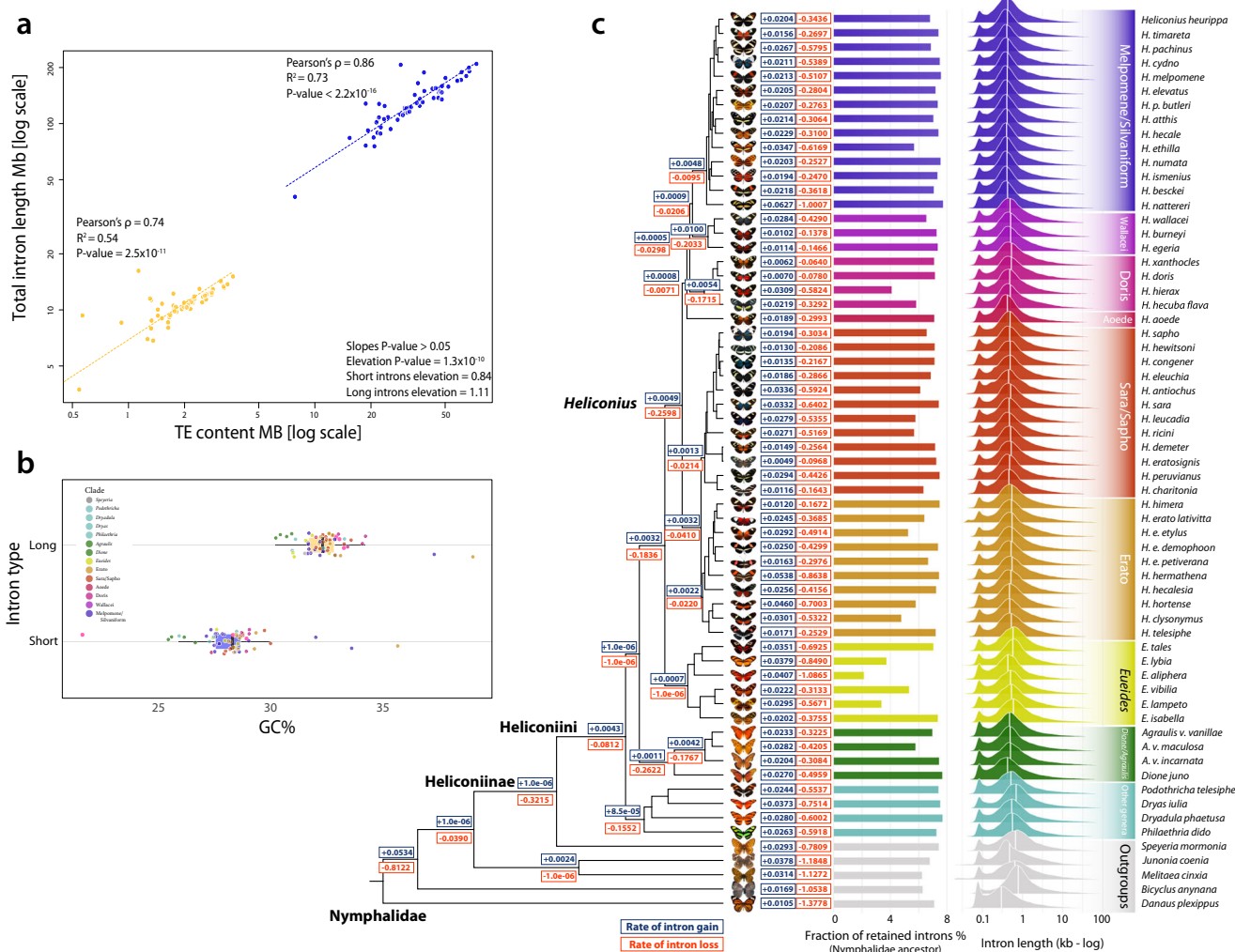

**Fig. 4 | Composition and intronic evolution. a** Log-log scatter plot showing high correlation between intron length and Transposable Element (TE) abundance. The significantly higher elevation of longer introns compared with short introns, indicates that large introns are more affected by TEs; short intron elevation = 0.84; $P$ value ≤ $2.5 × 10^{-11}$; long intron elevation = 1.11; $P$ value = $2.2 × 10^{-16}$ using standardized major axis (SMA) regression test $P$ value ≤ $1.3 × 10^{-10}$. **b** Box plot of GC content and intron size (One-sided Wilcoxon rank-sum test $P$ value < $2.2 × 10^{-16}$), bars indicate the interquartile ranges (IQR), while whiskers the quartile (Q) ± 1.5*IQR. **c** Rates of intron gain (blue) and loss (red) across Nymphalid phylogeny, and fraction of retained introns from the Nymphalid ancestor. On the far left, the log scale distributions of intron lengths, white vertical bars indicate the median values.

the effect of changing gene structure more than gene density (Supplementary Fig. 36). This suggests that selection may have favored a reduction in TEs in intergenic regions, perhaps to avoid the disruption of regulatory elements[19], consistent with TEs largely accumulating in the tails of chromosomes[20]. Although intron size varies significantly, the rate of gain/loss of introns, and the intron retention from the MRCA of Nymphalids, shows a relatively stable dynamic over the last 50 Mya in Heliconiinae, with no significant shift among species, and ~7% of ancestral intron sites retained across species (Fig. 4c). A similar pattern was reported in *Bombus*[21], but our results differ from drosophilids and anophelines, which show significantly higher intron turnover rates[22] (see Supplementary Note 4 for more details).

## Expansion and contraction of gene content
The Heliconiini tribe shows a diversity of key innovations in different aspects of their physiology and adaptation. These aposematic butterflies de novo biosynthesize their toxins when similar compounds are not available from their obligatory larval hostplant (Passifloraceae) for sequestration – a process called biochemical plasticity. These toxins not only make heliconiines distasteful to predators, but also play important role during mating[23,24]. The

Heliconiini also produce complex and diverse bouquets of pheromones[25], which can play an important role in speciation through the formation of pre-zygotic reproductive barriers, ultimately reducing gene flow and facilitating speciation. *Heliconius*, however, specifically show other traits, including an extended life-span and increased neural investment[26] compared with other butterflies and sister clades, which are thought to have evolved alongside pollen-feeding[10]. We tested if the origin of these suites of traits are associated with gene expiations/contraction at key points in the phylogeny, modeling the turnover rate of ortholog group (OG) size with CAFE v5[27] for 10,361 OGs using the 52 most complete genomes (BUSCO score ≥90%; Supplementary Fig. 37). The analysis identified 656 OGs that vary significantly in size across the phylogeny. The estimated gene turnover (λ) was of 0.006/gene gain-loss/Mya. This is relatively high compared with rates for *Bombus* (λ = 0.004) and anopheline species (λ = 0.003)[21,22], but similar to drosophilids (λ = 0.006)[28]. The base of the phylogeny showed relatively strong OG expansions, with few contractions, followed by stasis. While *Dione + Agraulis* and *Eueides* stems have similar proportions of expanded/contracted OGs, *Heliconius* shows 48 contracted OGs but only eight expanded OGs (Fig. 3c, Supplementary

Data 7) suggesting the phenotypic innovations that occurred in this branch were not due to widespread gene duplication.

Several OGs were identified to be expanded multiple times across the phylogeny and some of these may be directly associated to previously described key innovations/phenotypic traits across Heliconiini. For example, we find that cytochrome P450 (P450s) genes expanded in the common ancestor of the subfamily Heliconiinae, the tribe Heliconiini, the *Dione* + *Agraulis* stem, and within the genus *Heliconius* in the Erato group and Silvaniform/Melpomene stems. In insects, P450s play important roles in the detoxification of specialized metabolites, hormone biosynthesis/signaling, and biosynthesis of cyanogenic glucosides in heliconiine butterflies, which form the basis of their chemical defence[24]. Notably, a range of diet related OGs are also highlighted: Glucose transporters and Trypsins expanded several times in Heliconiinae, Heliconiini, *Eueides*, and the Silvaniform/Melpomene stem. Although glucose transporters play an important role in energetic metabolism in all animals, in phytophagous insects they are also hypothesized to be involved in the sequestration and detoxification of specialized metabolites from plants[29]. There are also expansions in Lipases enzymes, and OGs linked to in energetic metabolism and in pheromone biosynthesis in the Sara/Sapho + Erato stem and Silvaniform/Melpomene clades. At the stem *Heliconius* species there is one duplication of *methuselah*-like, a G-protein coupled receptor, involved in oxidative stress response, metabolic regulation, and lifespan[30], together with *Esterase P*, and a *juvenile hormone acid methyltransferase* (*jhamt*) which expanded three times, and the expansions of the single-copy Cuticle protein CPCFC (Supplementary Fig. 38). Taken together these expansions events offer good candidates for pathways which may be linked to the derived life history traits and chemical ecology of the Heliconiini.

We further expanded the previous unsupervised analysis by focusing on 57 gene families (GF), which includes a range of biological functions (Supplementary Data 8). We used measures of "phylogenetic instability" and the gene turnover rate ($\lambda$, CAFE), to explore their dynamics. The average instability score was 37.45 while the average $\lambda$ is 0.005, with the number of OGs per family positively correlated with $\lambda$ (Pearson's $\rho = 0.42$). Sodium/calcium exchanger proteins and the Hemocyanin superfamily show the highest instability and turnover rates (Supplementary Fig. 39). This analysis also identifies GFs which expanded in key periods of heliconiine diversification, including Hemocyanins, Lipases, Trypsins and Sugar transporters and the Major Facilitator Superfamily (Fig. 5a). The most notable are the P450 CYP303A1-like gene (Supplementary Fig. 40), a highly conserved protein in insects that has a pivotal role in embryonic development and adult molting[31], and two Hemocyanins, a hexamerin storage homolog expanded in the *Dione* + *Agraulis* + *Eueides* + *Heliconius* clade, and the arylphorin homolog, expanded in *Eueides* and *Heliconius*. These hexamerins function as storage proteins, providing amino acids and energy for non-feeding periods, such as molting and pupation, and may also transport hormones[32]. We further characterized OGs within each GF, aiming to test for correlated gene expansions/contractions and shifts in selective pressures between *Eueides* and *Heliconius* (see Supplementary Note 5 for more details), and found that Hemocyanins show, among several GFs, evidence of divergent selection regimes ($\omega$) between *Eueides* and *Heliconius*, alongside Trypsins, Protein kinases, P450s, Sugar transporters, Ion and ABC transporters (Fig. 5b). Curiously, a contraction in the Hemocyanin superfamily was only observed in *H. aoede*, in our data, the only *Heliconius* species that do not to feed on pollen, marking hexamerins a potential mechanistic link to the divergent strategies for nitrogen storage in pollen-feeding *Heliconius* (Fig. 5a). However, our analyses indicate that a number of putatively important changes in gene family size not only occurred at the stem of *Heliconius*, but also in more basal branches at level of subfamily and tribe, before the adaptive radiation of *Heliconius*.

## Selection across the Heliconiini radiation

Selection regimes shaping Heliconiinae coding genes were further investigated using the adaptive branch-site random effects likelihood (aBSREL) method. Again, we aimed to examine positive selection at the *Heliconius* stem, and contextualize these patterns by testing and measuring the degree of diversifying positive selection at more basal branches. First, when single-copy orthologous groups (scOGs) are classified according to their phylogenetic attribution (*i.e.*, where they appeared throughout the phylogeny), they show a trend towards increased purifying selection from young to older genes (Fig. 5c), suggesting that genes become more stable with time, probably reflecting increased functional importance. The signature of diversifying positive selection was assessed on five basal branches of the Heliconiinae phylogeny where key ecological transitions occur. From the Heliconiini to *Dione* + *Agraulis* + *Eueides* + *Heliconius*, *Eueides* + *Heliconius*, *Eueides*, to the *Heliconius* stem. Overall, the Heliconiini branch evolved under the strongest selection, followed by the *Eueides* and *Heliconius* branches, and finally by the *Eueides* + *Heliconius* branch (Fig. 5d). The number of genes with a signal of diversifying positive selection varies between branches, with the *Dione* + *Agraulis* + *Eueides* + *Heliconius* and *Heliconius* stems having the highest number of enriched biological processes (BPs), followed by Heliconiini and *Eueides* stem, and *Eueides* + *Heliconius*. A notably high proportion of branches are enriched for BPs relating to neuronal development and cellular functions, including the regulation of hippo signaling, stem cell differentiation and cell-cell adhesion, and genes associated with asymmetric division (Supplementary Data 9–11). Using a network-based approach, which integrates both primary and predicted interactions to predict gene function, we examined connections between selected genes. Although the amount of network interactions shows a significant degree of connectivity (absolute number of interactions) in the branches leading to *Dione* + *Agraulis* + *Eueides* + *Heliconius* (834 interactions), *Eueides* + *Heliconius* (627), *Eueides* (531), Heliconiini (410), and *Heliconius* (320) the network density shows a different picture, with *Heliconius* having a markedly higher density (the portion of the potential connections in a network that are actual connections) with -0.3 versus -0.2 for the other networks, in the case of BP networks (Supplementary Figs. 41 and 42, Supplementary Data 11, and Supplementary Note 6). The enriched molecular functions (MFs) in this densely connected *Heliconius* network are characterized by BPs related to response to DNA damage/repair, neuroblast division, and neural precursor cell proliferation, glial cell development, cell-cell junction assembly, asymmetric stem cell division. This concentration of neurogenesis-related functions differs from enrichment in other networks, which appear more variable. Finally, we note multiple genes that show a signature of diversifying positive selection on more than one branch. One of them is the *Notch* homolog, an essential signaling protein with major roles in developmental processes of the central and peripheral nervous system[33]. Notch regulates neuroblast self-renewal, identity and proliferation in larval brains, and is involved in the maintenance of type II neuroblast self-renewal and identity[34]. Overall, these findings support the idea that many genomic changes that can be putatively linked to key *Heliconius* traits reflect a continuation or exaggeration of changes that occur in earlier Heliconiini lineages, suggesting a more gradual pattern of genetic evolution that precedes the adaptive radiation of *Heliconius*.

## Acceleration of conserved non-exonic elements (CNEEs)

The scan for diversifying positive selection on protein-coding genes showed interesting patterns that could be correlated to the evolution of phenotypic traits in Heliconiini. However, as we have seen, selection on the stem of *Heliconius*, although strong, does not seem to affect a

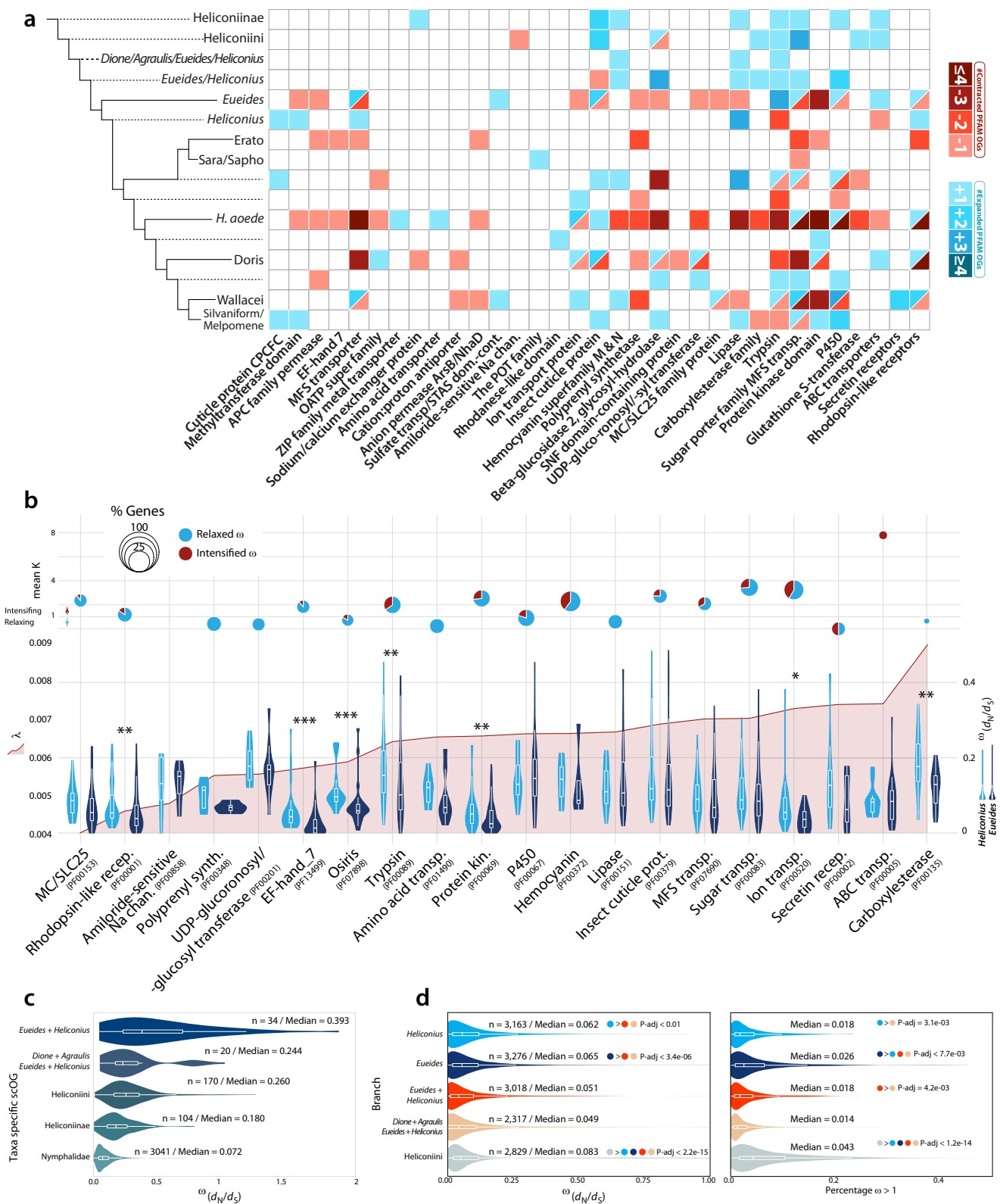

high number of genes. We therefore expanded our scope to non-coding regions, specifically to regions of the genome that are conserved across the phylogeny but show altered patterns of evolution on the *Heliconius* stem. Comparative genomics approaches have assumed a fundamental role in the identification of conserved and functionally important non-coding genomic regions[35–38]. One of the most prominent hypotheses is that these regions function as *cis*-regulatory elements (such as enhancers, repressors, and insulators) and determine

tissue-specific transcripts during developmental stages. To determine the extent of non-coding molecular evolution on the radiation of *Heliconius* butterflies, we compiled a total of 839k conserved elements (CEs) across the 63-way genome alignment (for a comparison, 1.95 M CEs were found in birds[39]), leveraging a statistically neutral substitutional model, which considers phylogenetic distances and species relationships, to provide a more rigorous measure of actual evolutionary constraint[40]. Of the total CEs, 473k (56%) overlap with protein

**Fig. 5 | Differential evolutionary rates gene families and scOGs across Heliconiini butterflies. a** Heatmap showing the different expansions and contractions in multiple gene families. Several gene families have been contracted in *H. aoede*. **b** Plots showing different evolutionary features of some of the analyzed gene families (minimum of 3 genes in *Eueides* spp. and *Heliconius* spp.). At the top section, dynamic pie charts showing mean $K$ value (selection intensifier parameter). Values below one indicates a relaxation, while above one indicates intensification towards diversifying positive selection. The size of the pie charts indicates the fraction of genes under intensification (red) and relaxation (blue), and it is scaled according to the proportion of genes for which $K$ was significantly different from $H_0$ (No difference) (see Methods). For different gene families the panel below shows the gene turnover rate ($\lambda$) (left $y$-axis); right $y$-axis shows the distributions of mean $\omega$ for near-scOGs (see Methods) in *Eueides* and *Heliconius* (right y-axis). Asterisks indicate significant shifts between *Eueides* and *Heliconius* (One-sided Wilcoxon rank-sum tests; *≤0.05; **≤0.01; ***≤0.001). **c** Violin plot showing the distributions of mean $\omega$ rates ($d_N/d_S$) in scOGs according to their lineage-specificity (one-sided Wilcoxon rank-sum test, $P$-value < 0.05). **d** Distribution of mean $\omega$ rates (left) for scOGs on six branches of Heliconiini, and the proportion of genes for which $\omega$ is higher than one (right) (one-sided Wilcoxon rank-sum test, $P$ adjusted values ≤ 0.05). For all plots boxplots within the violin plots bars indicate the interquartile ranges (IQR), while whiskers the quartile (Q) ± 1.5*IQR.

coding loci and 143k (30%) with coding exons, with 680k classified as conserved non-exonic elements (CNEEs), which were subsequently filtered (see Methods) to obtain a final set of 430,606 candidate CNEEs from the 63-way whole-genome alignment (811,696 in birds[39]); 202k intronic and 227k intergenic, for a total data set of 46,877,100 base pairs of aligned DNA.

We first checked for evidence of putative regulatory function by looking at the relationship between CNEEs and accessible chromatin, using ATAC (assay for transposase-accessible chromatin) peaks of 5th instar caterpillars from two tissues, brain and wing imaginal disc[41]. We found that in both tissues CNEEs overlap ATAC peaks twice as often as expected under a random distribution (permutation $P$-value < 0.0001), with brain tissue having a slightly higher increase of 2.4 fold-enrichment, compared with the imaginal disc tissue (2.0 fold-enrichment). This is in spite of imaginal discs having twice as many ATAC peaks, covering twice the genomic region. This indicates that our annotated CNEEs are consistent with being putative functional elements and suggests that regulatory regions associated with brain tissue may be under more constraint, with a more conserved regulatory architecture.

Because of the putative regulatory relevance of CNEEs we applied a Bayesian method[42,43] to detect changes in conservation of these elements at the stem of *Heliconius*, aiming to identify putative regulatory regions responsible for morphological and physiological adaptations of these butterflies. In total, we found that approximately half of the CNEEs (51%) experienced an acceleration in evolutionary rate at some point in the phylogeny. Around 95k elements experienced acceleration under a "full model" (M2), meaning that the latent conservation states **Z** (−1: missing, 0: neutral, 1: conserved, or 2: accelerated) can take any configuration across the phylogeny, while 122,445 elements best fit the lineage-specific model (accelerated on the *Heliconius* stem branch; M1), where substitution rates on the branches leading to target species are accelerated whereas all other branches must be in either the background or conserved state; of them 2,536 were accelerated (aCNEEs) at the stem of *Heliconius*. Among this list, we tested if there is enrichment of aCNEEs in accessible chromatin of brain and wing imaginal disc and found that in both tissues there was a similar fold-enrichment of 1.08 and 1.09, for brain and wing tissue, respectively ($P$-value = 0.04 for the brain tissue; $P$-value < 0.001 for imaginal disc). We then checked for enrichment of aCNEEs across genes, as well as their spatial distribution across the genome to identify genes most affected by the acceleration, or large regulatory hubs. We found 37 genes that harbor more aCNEEs in their putative regulatory domains than expected by chance (Supplementary Data 12). Among them, there are multiple genes linked to axon pathfinding[44,45] (two genes homologous to *Uncoordinated 115a, Unc-115a, Eisa2300G23*: 5 aCNEEs; *Eisa2300G24*: 6 aCNEEs; adj. $P$-value < 0.026; *Multiplexin, Mp, Eisa1200G485*: 5 aCNEEs; adj. $P$-value < 0.026), synaptic pruning and transmission[46], and long term memory[47] (*Beaten path Ia, beat-Ia, Eisa2300G476*: 3 aCNEE; adj. $P$-value = 0.022; *Tomosyn, Eisa1400G28*: 4 aCNEEs; adj. $P$-value = 0.026). We also find examples such as *Nicastrin* (*nct*), which encodes a transmembrane protein and a ligand for Notch (N) receptor (*Eisa0300G576*: 3 aCNEEs; adj. $P$-value = 0.0014), and is required for neuronal survival during aging and normal lifespan,

functioning together with a *Presenilin*-homolog (*Psn*)[48] (*Eisa1800G396*: 1 aCNEE) which, although not enriched, also has one aCNEE in its regulatory domain. Finally, two pheromone binding proteins (*PhBPloc02ABP1*: 2 aCNEEs; adj. $P$-value = 0.026; *PhBPloc08ABPX*: 3 aCNEEs; adj. $P$-value < 0.05) and a sugar taste gustatory receptor (*Eisa0300G244*: 3 aCNEEs; adj. $P$-value = 0.041) are also highlighted as having multiple aCNEEs in their regulatory domain on the stem *Heliconius* branch.

The spatial enrichment analysis also highlighted 55 genomic regions significantly enriched upon $P$-value correction (Supplementary Data 13). Two of these correspond to a 150 kb (8 aCNEEs) and 120 kb (4 aCNEEs) gene deserts, meaning they contain no annotated protein coding gene. In proximity of these regions are mainly coding transposable elements, or viral ORFs, such as x-elements or retrovirus-related Pol polyproteins of *Drosophila*, and nearby *collagen alpha-1(III) chain-like*, *Argonaute 2*, *Osiris 21* (*osi21*) and *spalt major* (*salm; Eisa0200G420*: 8 aCNEEs), an important zinc finger transcriptional repressor that mediates most decapentaplegic (dpp) functions during the development of the wings. The product of *salm* is also required for cell specification during the development of the nervous system, muscle, eye or trachea[49]. Together with the notion that gene deserts have pivotal regulatory functions[50], this makes these two regions important candidate regulatory hub for developmental processes in *Heliconius*. A further enriched genomic region is located on chromosome 20 (Fig. 6). This region harbors eight aCNEEs distributed across two putative regulatory domains of two genes, both homologs of *osa*, which encodes for a subunit of the Brahma-associated protein (BAP) chromatin remodeling complex, part of the SWI/SNF chromatin-remodeling complexes. This complex functions to alter the accessibility of transcription factors to genomic loci. As such, it plays important gene regulatory roles in multiple contexts[51]. In *Drosophila*, it controls escorting cell characteristics and germline lineage differentiation[52], but the complex is also implicated in inducing the transcription of *crumbs* (*crb*), which we also found to have one aCNEEs in its putative regulatory domain. Crumbs, in turn, is a transmembrane protein which negatively regulates the Hippo signaling cascade, and plays an integral role in cell proliferation and tissue growth regulation[51]. Additionally, the silencing (by RNAi) of different subunits of the BAP complex results in disrupted short- and long- term memory, while direct silencing of *osa* impaired the retention of long-term memory[53]. Given that long-term memory is thought to be stable across longer periods in *Heliconius* than related genera[26], these reflect clear candidate loci of interest. We also examined evidence of GO term functional enrichment among the 2536 *Heliconius*-specific aCNEEs (Supplementary Figs. 43–45), using different approaches which resulted in similar enriched categories (Supplementary Data 14). Specifically, 36 aCNEEs are linked to strongly enriched transcription factors and receptors related to imaginal disc-derived wing morphogenesis (e.g.: *dl, osa, ser, lgs, dll, fz2, sfl*), and retinal cell differentiation (e.g.: *salm, emc*), 36 aCNEEs near 14 genes are related to the Notch signaling pathway (e.g.: *agxt, ham, got1, nct, psn, noc, wry, nedd4*), and 20 aCNEEs near 11 genes are related to feeding behavior (e.g.: *for, 5-ht2a, dip-kappa*).

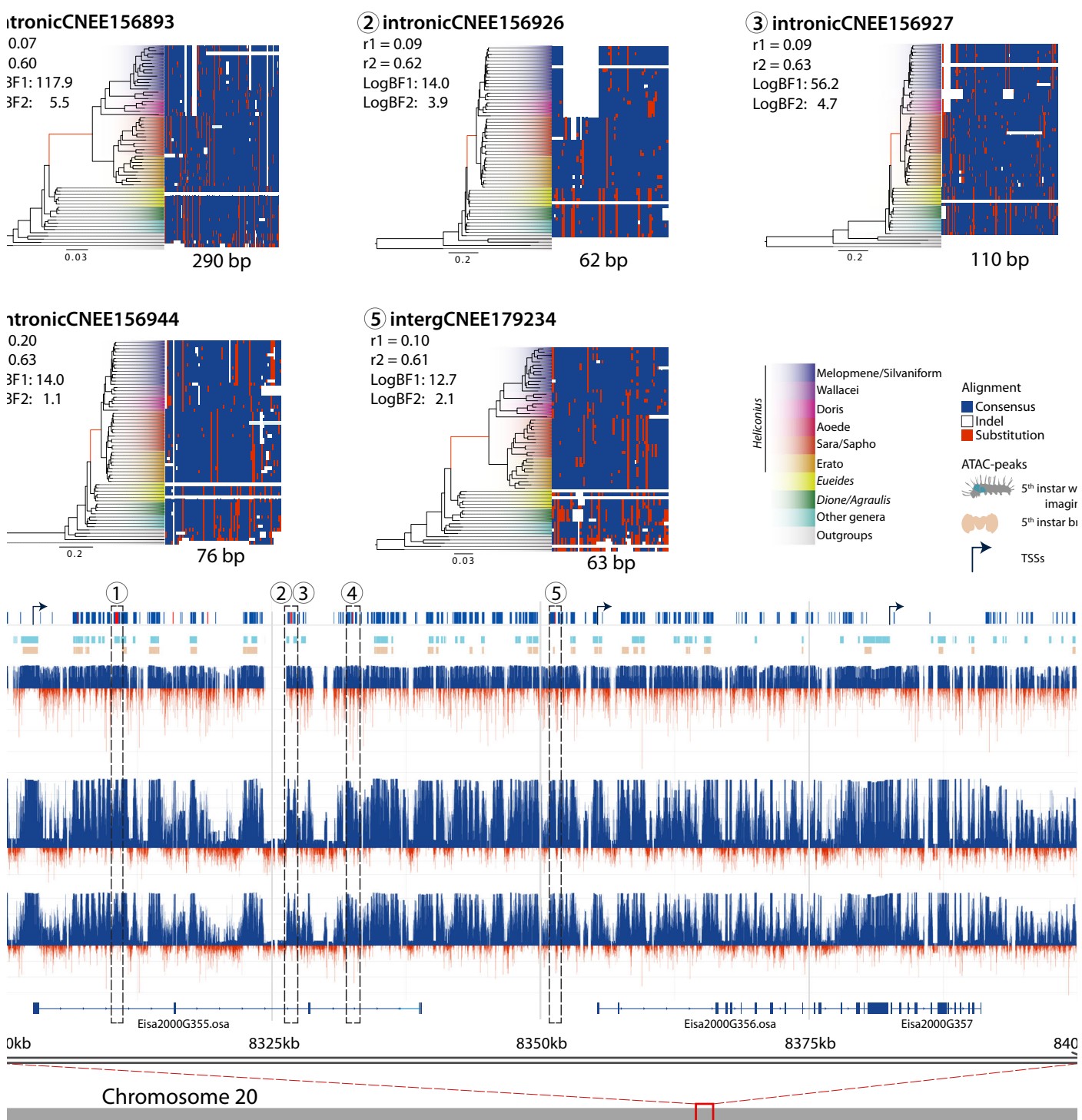

**Fig. 6 | Chromosome 20 enriched genomic window.** Diagram showing the distribution of CNEEs in one of 100 kb enriched window across the reference genome of *E. isabella*. From the bottom-up, the figure shows three genes, two of them homologs of the *Drosophila osa*. Above that are the CONACC scores obtained from the full alignment (63 species), for only the non-*Heliconius* species, and only the *Heliconius* species. In red the negative values indicate the acceleration of a given position of the alignment, and in blue the positive values indicate conservation. Above that are the ATAC peak distributions of two tissues from 5th larva instar, brain (in brown) and imaginal disc (in aquamarine), shown alongside the distribution of CNEEs in the region in dark blue with the eight aCNEEs. Numbers indicate the five aCNEEs selected as examples of the *Heliconius* aCNEEs, which are expanded at the top of the figure. In these examples, the alignments show conserved (nucleotides similar to the consensus, in blue) and accelerated sequences (nucleotides that differ from the consensus, in red). For each of the five aCNEEs the species phylogeny of the Nymphalids is shown where the branch lengths indicate the acceleration of the evolutionary rate for each given aCNEE. The branch that corresponds to the *Heliconius* stem is in red. For each aCNEE the two log-BFs and conserved ($r_1$) and accelerated rate ($r_2$).

## Candidate genes for derived traits of *Heliconius*

Within the Heliconiinae, *Heliconius* display a number of divergent traits and innovations[10]. Here, we highlight how our results reveal new biological insights into these traits, focusing on two case studies; changes in neural composition in Heliconiini, and the enzymatic processes associated with breaking down pollen walls to aid their digestion during pollen feeding. These two examples, illustrate the potential of large, densely sampled genomic datasets to both generate and test

adaptive gene-phenotype hypotheses, using both unguided and more targeted analyses.

Within in central brain, mushroom bodies are paired organs that receive visual and/or olfactory information, and play a pivotal role in learning and memory[54]. These structures show huge variation across Heliconiini, but a particularly large expansion occurred at the *Heliconius* stem, where mushroom body volume and neuron number more increased by several-fold, accompanied by a major shift towards increased dedication to processing visual information[11,26]. These changes are accompanied by enhanced learning and memory performance[26], and likely facilitate the foraging strategies deployed during pollen feeding. However, the molecular mechanisms underpinning these events – or, indeed, any case of mushroom body, or brain expansion in insects – are unknown. Given the lack of variation in closely related species suitable for alternative approaches, comparative genomics reflects the best route to identifying genes linked to this shift in brain morphology. Our selection analyses highlight pathways that could regulate neural proliferation. These include the Hippo signaling pathway, which regulates cell growth and proliferation of neural stem cells and neuroblast quiescence[55]. Multiple and repeated signs of diversifying selection are identified on genes related to the Hippo signaling pathway, including *Focal adhesion kinase* (*Fak*), *lethal (2) giant larvae* (*lgl*), *Sarcolemma associated protein* (*Slmap*), and *Akt kinase* (*Akt*), on the *Dione + Agraulis + Eueides + Heliconius* stem, which regulate cell polarity, asymmetric division and cell proliferation[56], and two other genes, *Moesin* (*Moe*) and *F-box and leucine-rich repeat protein 7* (*Fbxl7*), in the *Eueides + Heliconius* stem. Moe drives cortical remodeling of dividing neuroblasts[57], while Fbxl7 affects Hippo signaling pathway activity[58]. Finally, *Ctr9*, *dachsous* (*ds*), *falafel* (*flfl*), and locomotion defects (loco) were identified at the *Heliconius* stem (Supplementary Data 9). Ctr9 is involved in the proliferation and differentiation of the central nervous system[59], Flfl is required for asymmetric division of neuroblasts, cell polarity and neurogenesis in mushroom bodies[60], Ds is a cadherin that interacts with the Hippo signaling pathway[61], and loco is an activator of glial cell fate, essential cells in efficiently operating nervous systems[62]. Similarly, our analysis of conserved non-coding elements reveals multiple loci nearby genes with known roles in neural development, synaptic pruning, and long-term memory. Collectively, these provide the first candidate loci linked to mushroom body expansion in any insect and provide ample gene-phenotype hypotheses for further investigation.

Despite being an evolutionary key innovation in *Heliconius*, similarly little is known about the mechanism underpinning pollen-feeding itself. Saliva probably has an important role in the external, enzymatic digestion of the pollen wall[10]. The leading candidates for these enzymes are serine proteases, homologs of the silkworm cocoonase[63,64], which digests the cocoon during eclosion[64]. Because butterflies do not produce a cocoon, it has been proposed that the duplications of cocoonase orthologs may have been co-opted to digest pollen[63]. Given our order of magnitude larger sample, and having not highlighted this gene family in our unguided analysis, we re-evaluated the evolution of these genes by reassessing the evolutionary history of this gene family, and evidence of gain-of-function. We identified 233 cocoonase loci (Supplementary Data 15) across all Heliconiinae and found that the duplications not only predate the split between *Heliconius* and *Eueides*, but affect the whole Heliconiini tribe and its outgroup *S. mormonia* (Fig. 7a). All species have at least four copies, located at the minus strand of chromosome 15 with remarkable conserved synteny (Fig. 7b and Supplementary Data 16). Substrate, cleavage, and active sites of the functional domain show very high conservation throughout the dataset. Three independent tandem duplications from the same original copy are very likely responsible of the emergence of *Coc1A*, *Coc1B*, *Coc2* and *Coc3* (Fig. 7a). High level of purifying selection is detected across the four OGs (Fig. 7a). A scan of all internal branches for signs of diversifying positive selection shows

that the branches of *D. juno Coc2* in-paralogs; the stems of all *Heliconius Coc1A* and *Coc1B*, and the two branches of the Silvaniform/Melpomene *Coc2* out-paralog, show signs of positive selection. Two of these events involve loci from the non-pollen feeding *H. aoede*. We therefore tested if these loci show signs of relaxation in this species. Surprisingly, while no significant differences were detected for *Coc1B*, an intensification of selection was detected for *Coc1A* ($K = 1.41$; $P$-value = 0.001). To gain more insight into a gain-of-function hypothesis, we modeled the 3D structure of the full-length protein sequences, a trypsin-like serine protease composed of two folded beta barrels connected by a long loop positioned at the back of the active cleft[63], and, by adopting a graph-based theory approach (Supplementary Fig. 46), we inferred the key residues driving the structural differences among loci. Notably, the methodology clustered all the structures into four groups, consistent with the phylogenetic analysis, plus a fifth group for the Melpomene/Silvaniform *Coc2* sub-clade, which evolved under diversifying positive selection. We found that seven residues drive the overall clustering (Supplementary Figs. 47 and 48), and these lie in three regions of the 3D structures, corresponding to three loops in regions highly exposed to the solvent (Fig. 7d). The structural alignment of the predicted cocoonases with the X-Ray structures of several homologous human serine proteases (Supplementary Fig. 49), shows that the two largest loops (pos: 217-217 and pos: 119-122) corresponds to highly flexible regions in the experimental structures (i.e., B-factor), in contrast with the shorter third loop (pos: 68-71), which is in turn analogous to a region with higher stability. These analyses suggest that the duplicated genes might have gained the capacity to bind and process different substrates by changing their flexibility throughout the radiation of Heliconiini. This is consistent with the hypothesis that, in order to obtain a gain-of-function and to give rise to new interactions, a protein needs to change few sites in intrinsically disordered regions[65]. Our combined results present a more complex story than previously described, and both the high copy number variation and patterns of selection within Heliconiinae appear inconsistent with these genes playing a critical role in the evolution of pollen feeding (see Supplementary Note 7 for more details).

In conclusion, we have curated available genomic data and reference genomes to build a tribe-wide dataset for Heliconiini butterflies. Using the resulting phylogenetic framework, we examined patterns of genomic change at points in the species tree around which key phenotypic innovations are expected. We investigated the evolution of genome size, its effect on protein-coding gene expansions and contraction, and selective forces such as diversifying positive selection on protein coding genes and the acceleration of conserved non-coding genes. Supported by the characterization of all these genomic features, our analyses ultimately allowed us to narrow down candidate genes that could be further tested to explore more in depth the molecular architecture of key innovations in this enigmatic group of butterflies. This provides a genome-wide perspective of the strong but gradual selection events that occurred at the basal branches of the Heliconiini tribe, exemplified by expansions in gene families and OGs linked to biochemical processes relevant to cyanogenic defences, dietary shifts, and longevity, with signatures of adaptive evolution. Notably, multiple strands of evidence implicate selection acting on both coding and non-coding loci affecting neural development and proliferation, synaptic processes, and long-term memory, in line with evidence of substantial variation in the structure of Heliconiini brains[10,11,26]. These results highlight how individual loci, as well as wider pathways, such as the Notch and Hippo pathways, might have evolved under a strong diversifying selection, providing the first gene-phenotype links underpinning mushroom body expansion[26]. Finally, our test for acceleration of putative *cis*-regulatory elements (CNEEs) at the stem of *Heliconius* identified more prevalent positive selection on non-coding elements compared to protein coding genes at the origin of *Heliconius*. This suggests the suite of derived phenotypes in this

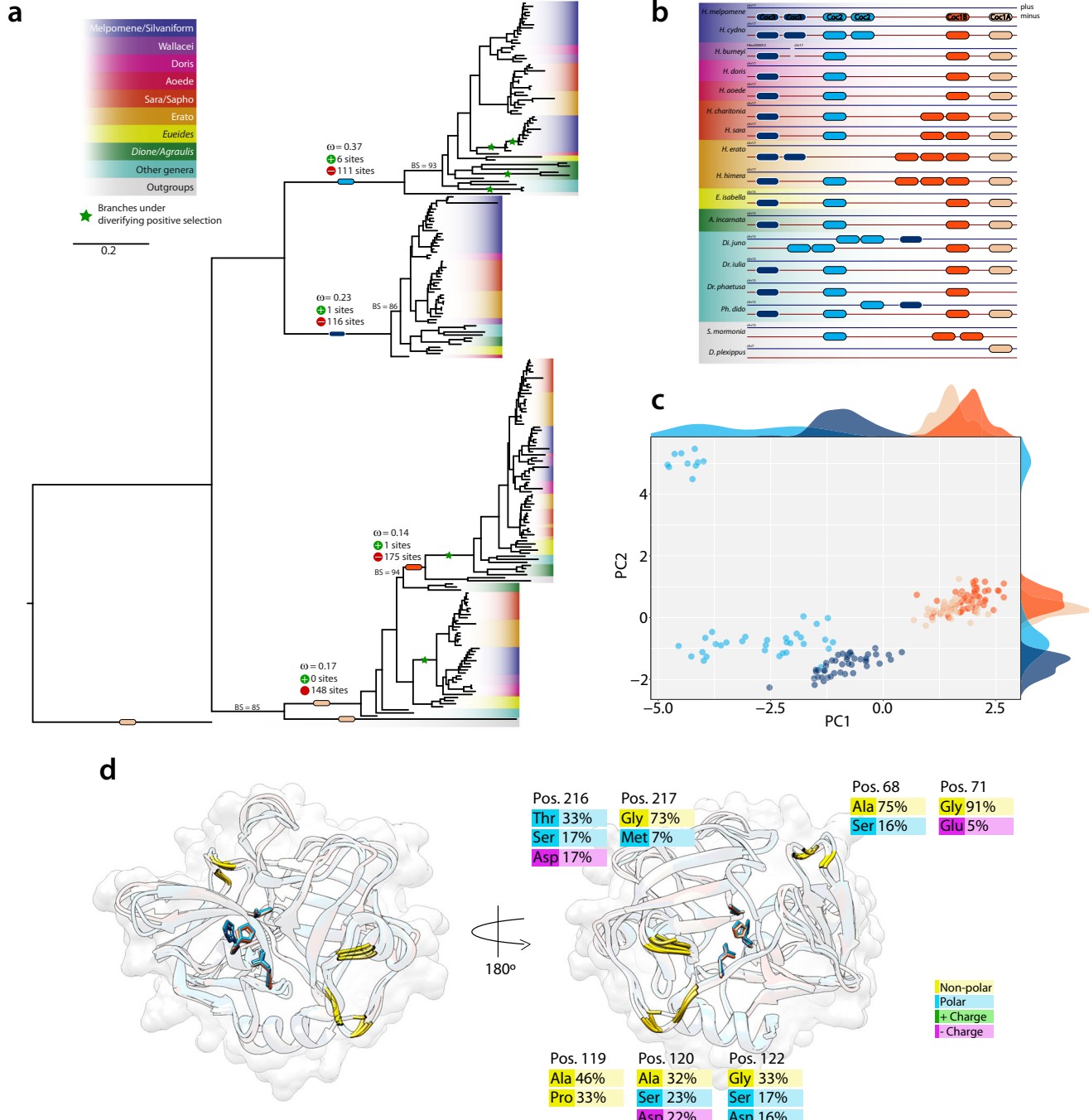

**Fig. 7 | Cocoonase evolution and structural divergence across Heliconiini.**
**a** Maximum likelihood phylogeny of nucleotide sequences of the four cocoonase loci across Heliconiinae. Colored blocks on branches show the stem for each locus. The green star indicates branch under diversifying positive selection. Bootstrap (BS) values for main branches are listed for those with values below 95. **b** Synteny map of the different loci for the genomes with highest contiguity. **c** PCA of the network-based analysis of 181 predicted protein models. **d** Structural alignment of

the closest sequence to the centroid of each of the five clusters of the PCA (colored ribbons). For each of structure the three active sites of the active cleft are depicted as sticks, while in gold the seven identified positions the best explains the clustering of the PCA. On the left the same structure rotated 180°. For each of the positions the most frequent amino acids are shown with their respective frequency in the alignment.

genus might have largely evolved through changes in gene expression via modification of regulatory elements (e.g.: promoters, enhancers, and silencers)[41,66]. In conclusion, our work offers a comprehensive view to the evolutionary history of an enigmatic tribe of butterflies, the evolution of their genomic architectures, and provides the most thorough analysis of potential molecular changes linked to the physiological and behavioral innovations of a diverse group of butterflies. These gene-phenotype hypothesis, alongside our comprehensive

dataset, provide new opportunities to test and derive causative links between molecular and trait innovations.

## Methods

### DNA and RNA extraction and sequencing

Individuals of *Dryadula phaetusa, Dione juno, Agraulis vanilla vanillae*, were collected from partially inbred commercial stocks (Costa Rica Entomological Supplies, Alajuela, Costa Rica); while individuals of

*Agraulis vanilla incarnata* ([www.shadyoakbutterflyfarm.com](www.shadyoakbutterflyfarm.com)), *Speyeria mormonia washingtonia* (Washington, USA), *Philaethria dido* (Gamboa, Panama), *Podothricha telesiphe* (Cocachimba, Peru), *H. aoede* (Tarapoto, Peru), *H. doris* (Gamboa, Panama), and *H. cydno*, were collected from the wild, stored at first in RNAlater solution and subsequently kept at −80 °C. Samples collected in Peru were obtained under permits 0289-2014-MINAGRI-DGFFS/DGEFFS, 020-014/GRSM/PEHCBM/DMA/ACR-CE, 040−2015/GRSM/PEHCBM/DMA/ACR-CE, granted to Dr Neil Rosser, and samples from Panama were collected under permits SEX/A-3-12, SE/A-7-13 and SE/AP-14-18. High-quality, high-molecular-weight genomic DNA was extracted from pupae (commercial stock specimens) and adults (wild caught specimens), dissecting up to 100 mg of tissue, mainly from the thorax. Samples were snap frozen in liquid nitrogen and homogenized in 9.2 ml buffer G2 (Qiagen Midi Prep Kit) adding 19 µl of RNase A. The samples were then transferred to a 15 ml tube adding 0.2 µl of Protease K and incubated at ~50 °C for 2 h. Samples were transferred to Genomic columns and processed with a Qiagen Midi Prep Kit (Qiagen, Valencia, CA) following the manufacturer's instructions. DNA was then precipitated using 2 ml 70% EtOH and dissolved in water.

From the same stocks, RNA was extracted separately from six adult tissue (four wings; three heads; four antenna, legs and mouth parts; thorax; abdomen segments 1–3, abdomen 4–6), and five tissue parts from early ommochrome stage pupae (head and mouth parts; wings, antenna and legs; thorax; abdomen 1–3, abdomen 4–6). Each tissue was frozen in liquid nitrogen and quickly homogenized in 500 µl Trizol, adding the remaining 500 µl Trizol at the end of the homogenization. Phase separation was performed by adding 200 µl of cold chloroform. The upper phase was then transferred to RNeasy Mini spin column and processed with a Qiagen RNeasy Mini Prep Kit (Qiagen, Valencia, CA), before DNase purification using the Turbo DNA-free kit (Life Technologies, Carlsbad, CA) following the manufacturer's instructions. All the extractions were finally pooled keeping the same final RNA concentration from all samples.

Due to different DNA quality, different 3rd generation sequencing technologies were applied. Pacific Bioscience Sequel II / HiFi was used to sequence the *S. mormonia washingtonia* genome at the Genomics Resource Center (GRC), University of Maryland Baltimore (UMD-IGS). Other Pacific Bioscience data were generated from *Dr. phaetusa*, *Di. juno*, and *A. vanillae* (Florida) at the Center for Genomic Research, University of Liverpool using PacBio sequel SMRT cell (2.0 chemistry), adding a library of Illumina DNAseq (HiSeq2500 150 × 2) data for *D. phaetusa*. For *Ph. dido*, *Po. telesiphe*, *A. vanilla* (Costa Rica), *H. aoede*, and *H. doris*, the 10x Chromium Library Prep was adopted and Illumina sequencing using 150 bp paired-end reads with NovaSeq FC S2, generating ~40 Gbp per species, and adding Nanopore 1D (up to 5 Gbp per flowcell) sequencing for *Ph. dido*, *Po. telesiphe*, *A. vanillae* (Costa Rica), both performed at the Institute of Applied Genomics (IGA), Udine, Italy. Polyadenilated Illumina RNAseq data (125 bp × 2) were generated for *Dr. phaetusa and Di. juno* at the Institute of Applied Genomics (IGA), Udine, Italy.

Short-read data for *A. vanillae* (Peru; ERR5235460), *E. lampeto* (ERR5235459), *E. vibilia* (ERR5235454), *E. aliphera* (ERR5235452), *E. lybia* (ERR5235468), *E. tales* (SRR8883890), *H. telesiphe* (SRR8883900), *H. clysonymus* (SRR4032079), *H. hortense* (SRR4032054), *H. hecalesia* (SRR8883898), *H. erato petiverana* (SRR4032055), *H. erato etylus* (ERR5235453), *H. peruvianus* (ERR5235458), H. eratosignis (ERR5235467), *H. demeter* (SRR8883893), *H. ricini* (SRR4032011), *H. leucadia* (ERR5235456), *H. antiochus* (ERR5235455), *H. eleuchia* (SRR3102171), *H. congener* (SRR3102172), *H. hewitsoni* (SRR3102337), *H. sapho* (ERR266262), *H. hecuba flava* (ERR1143583), *H. hierax* (ERR1143585), *H. xanthocles* (ERR1143626), *H. egeria* (ERR5235461), *H. burneyi* (SRR8883892) *H. wallacei* (ERR1143625), *H. besckei* (SRR8883889), *H. ismenius* (ERR1143586), *H. numata* (SRR8883908), *H. ethilla* (ERR260305), *H. hecale* (SRR8883896), *H. atthis* (ERR5235451),

*H. pardalinus butleri* (SRR8883891), *H. elevatus* (SRR8883894), *H. pachinus* (ERS977714), *H. timareta* (SRR3102172), and *H. heurippa* (ERR3653294), were downloaded from NCBI and used for de novo assembly and/or to curate already available assemblies (see Supplementary Data 1 for details).

## PacBio genome assembly

PacBio Hifi CCS reads were directly used to assemble the genome of *Speyeria mormonia washingtonia* (Washington, USA) using HIFIASM v0.12-r304 [settings: −f0][67]; while for other species PacBio reads were corrected, trimmed and assembled using CANU v1.8 + 356 changes[68] [settings: genomeSize = 400 m; corMhapSensitivity = normal; corOut Coverage = 100; correctedErrorRate = 0.105; ovlMerThreshold = 500; batOptions = -dg 3 -db 3 -dr 1 -ca 500 -cp 50], as in Cicconardi et al. (2021)[20]. The resulting raw assemblies were subsequently corrected by remapping all uncorrected raw PacBio reads with PBMM2 v1.0.0 and ARROW v2.3.3 ([https://github.com/PacificBiosciences/GenomicConsensus](https://github.com/PacificBiosciences/GenomicConsensus)) was used to correct the assembly with three iterations. To further error correct contigs, short Illumina reads were mapped with STAR v2.7.2c[69], and PILON v1.23[70] was run for five iterations. Assemblies were then processed with PURGE HAPLOTIGS v20191008[71] using ad hoc -a parameter to optimize the removal of haplocontigs, artificially duplicated genomic regions due to heterozygosity. To correct for mis-assemblies POLAR STAR ([https://github.com/phasegenomics/polar_star](https://github.com/phasegenomics/polar_star)) was employed. This tool calculates read depth of aligned PacBio reads to the assembly at each base, smoothed in a 100 bp sliding window, merging regions of high, low, and normal depth. Low read depth outliers are identified, and contigs are broken at each such location. Contigs were then rescaffolded using P_RNA_SCAFFOLDER[72], which uses information from RNAseq mapping (if RNAseq were available), and LRSCAF v1.1.5[73], which uses information from long-reads. The resulting gaps were then filled using PacBio reads applying LR_GAPCLOSER v.1.1[74]. After the introduction of this PacBio information, we repeated the previous polishing procedure using five iterations of Pilon, plus three more iterations with Illumina RNA-seq data to correct indels only. Before the chromosome-level scaffolding, we used synteny maps implemented with BLAST[75] and ALLMAPS[76] to identify duplicated regions at the end of scaffolds, manually curating the scaffolds to trim them away.

## 10X genomics linked-read genome assembly

Sequenced Illumina paired-end reads from 10X Genomics libraries were input to the SUPERNOVA v2.1.1 assembler (10x Genomics, San Francisco, CA, USA)[77] for de novo genome assembly. No trimming was needed according to the assembler documentation. The assembly was started under the "supernova run [settings: --bcfrac=1]" module, and because each genome had a different DNA starting quality and genome size, the best amount of reads (--maxreads) was searched for and adopted to maximize contiguity, duplication level and completeness, executing BUSCO (Benchmarking Universal Single-Copy Orthologs; v3.1.0, Insecta_odb9)[78] at each test run. After the optimization the fasta file for the generated assembly was produced with "supernova mkoutput [settings: --style=pseudohap2 --headers=full --min-size=1000]". Subsequently, assemblies were processed with PURGE HAPLOTIGS using ad hoc -a parameter to remove haplocontigs. TIGMINT v1.1.2[79] with default settings was then adopted to identify misassemblies, and break the assembly, before the scaffolding procedures. If RNA-seq where available, P_RNA_SCAFFOLDER was used, adding LRSCAF using Nanopore data, if available, and a finale step using ARCS v1.1.0[80], a scaffolding procedure that utilizes the barcoding information contained in linked-reads to further organize assemblies. If Nanopore data were available LR_GAPCLOSER was applied to the final gap closure, using ARROW and PILON, as before, to correct the scaffolded assembly. The synteny maps implemented for the PacBio data were again implemented here to remove duplicated artifacts.

**Reference-based genome assembly**

To assemble the genomes from the retrieved Illumina PE reads from NCBI (see Supplementary Data 1) we implemented a reference-guided assembly approach, adapting and extending the protocol from Lischer and Shimizu[81]. The strategy involves first mapping reads against a reference genome of a related species (see Supplementary Data 1 'Ref Genome' field) in order to reduce the complexity of the de novo assembly within continuous covered regions, then integrating reads with no similarity to the related genome in a further step.

Modifications from the reference guided de novo pipeline[81] started in the 1st step, were paired-end illumina reads were mapped onto the reference genome with MINIMAP2 v2.17-r974-dirty[82], followed by PILON[70] [--changes --diploid --nostrays --fix all]. This added step improved the mapability of the target species to the reference by modifying the reference assembly at the nucleotide level. This step was repeated iteratively five times, until a plateau of mapped reads was observed. In the 2nd step, paired-end reads are mapped against the modified version of the reference genome using the fast-local mode of BOWTIE2 v2.2.1[83]. Reads were then assigned into blocks, defined as a region with continuous read coverage; and extended if regions were spanned by at least 10 proper read pairs. Subsequently superblocks were defined based on the non-overlapping blocks: consisting of a combination of two or more blocks until a total length of at least 12 kb was reached. Superblocks longer than 100 kb were split into several superblocks of 100 kb or less, overlapping by 300 bp (more details in Lischer and Shimizu 2017). Reads mapped to each superblock, and all unmapped reads with a mate mapped to the same region, were extracted using SAMTOOLS v1.14[84]. These reads were then separately de novo assembled using SPADES v3.15[85] in the 3rd step [--careful -m 250]. In this step all unmapped reads were de novo assembled with SPADES to integrate more diverged genomic regions. The resulting redundancy generated by superblock was therefore removed in the 4th step by assembling these contigs with the homology guided Sanger assembler AMOSCMP v3.1.0[86,87] using the same modified reference genome with default parameters. Because AMOSCMP does not return unassembled contigs, in the 5th step reads were realigned back to the supercontigs using BOWTIE2 [--sensitive], and all unmapped reads were de novo reassembled. The resulting contigs added to the list of supercontigs. In the 6th step supercontigs were validated and corrected by aligning all paired-end reads against them, filtering out reads with a mapping quality lower than 10. Additionally, a local realignment of reads around indels was performed using GATK v3.1[88] and PICARD v2.26[89], and misassemblies corrected using SAMTOOLS and BCFTOOLS v0.1.19[84]. In a 7th, and final step, reads were used to attempt scaffolding and gap closing with SOAPDENOVO2 VR240[90], and scaffolds shorter than 1 kb were discarded.

Extra scaffolding procedures were implemented to improve the previous reference guided de novo assembly pipeline[81]. Leveraging the very small genetic differences between these species and their reference genomes, RNA-seq data from these species, when possible (see Supplementary Data 1) were downloaded from NCBI concatenated, corrected and normalized using BBMAP v38.79[91] [target = 20 maxdepth = 20 mindepth = 5]. These reads were mapped using HISAT2 v2.1.0[92], and P_RNA_SCAFFOLDER[72], a genome scaffolding tool that searches "guide" pairs where two paired reads were mapped to two different contigs, to "guide" orientation and order the contigs into longer scaffolds. Following this step, we applied RAGOO[93] [-T sr], a homology-based scaffolding and misassembly correction pipeline. RAGOO[93] identifies structural variants and sequencing gaps, to accurately orders and orient de novo genome assemblies. ABYSS-SEALER v2.2.2 from the ABYSS package[94] was use as last step using multiple kmers [-k99 -k97 -k95 -k93 -k91 -k89 -k85 -k81 -k77 -k73 -k69 -k65 -k61 -k57] to finalize the assembly and attempt to close remaining gaps.

**Curation of available illumina assemblies**

We took advantage of the available Heliconiini "Discovar" assemblies[7], including them in our dataset, but before proceeding with downstream analyses (e.g., gene annotation), a small, but effective curation was applied. At first, we checked for contaminants (see below), and then checked for completeness (using BUSCO, see above), the presence of haplocontigs. Raw Illumina reads were remapped onto their own assembly and PURGE HAPLOTIGS, with ad hoc -a parameter, was adopted to remove them, followed by a scaffolding procedure with SOAPDENOVO2 (127mer), and further synteny mapping implemented with ALLMAPS, using their closest available relative with a reference assembly. This procedure was adopted to maximize the contiguity. For future studies it should be noted that this may inhibit further insight into different genomic rearrangements from the reference species by assuming conserved synteny, a procedure similar to this was recently adopted to scaffolded draft genomes[95]. Finally ABYSS-SEALER v2.2.2 from the ABYSS package[94] was used for gap filling (see above).

**Bacterial contamination & assembly completeness assessment**

After the genome assembly stage all datasets were analyzed to remove contaminants. BLOBTOOLS v1.1.1[96] was implemented using BLASTN [-evalue 1e-25 -max_target_seqs 1] and the NCBI nucleotide collection (#seqs: 49,266,009, retrieved September 2018). Taxon identities for each hit were retrieved and we filtered out any scaffold and contig that were assigned to fungal and bacterial contaminants. Furthermore, mitochondrial sequences were identified by blasting (BLASTN) contigs and scaffolds against the mitochondrial genome of available Heliconius ssp. (HE579083.1, NC_026564.1, NC_026463.1, KP281778.1, KP294327.1, KP100653.1, NC_024744.1, NC_024741.1, KM068091.1, KM014809.1, NC_027516.1, KP784455.1, NC_024864.1, KM208636.1). A combination of BUSCO v3.1.0 (Benchmarking Universal Single-Copy Orthologs)[78] with the Insecta set in ORTHODB v.9 (odb9 1958 genes) implemented using default parameters [-m genome], and EXONERATE v2.46.2[97], using missing and fragmented amino acid sequences, was used to assess genome completeness and duplicated content.

**Whole genome alignment**

Prior to the transcriptome annotation, the complete single-copy orthologous genes identified using BUSCO were used to generate a first draft of the phylogeny to guide the whole genome alignment. From each locus the nucleotide sequence (nt) was extracted and aligned separately with MACSE v2.03[98], and subsequently concatenated into a single alignment. A maximum-likelihood (ML) search was adopted to estimate the phylogenetic tree as implemented in FASTTREE v2.1.11 SSE3[99]. All 63 Nymphalid genomes were soft-masked for repeats as described above, and CACTUS[100,101] was run using the guide-phylogeny, with genomes at chromosome level set as reference. Post-processing was performed by extracting information from the resulting hierarchical alignment (HAL). We then used HALSUMMARIZEMUTATIONS, from the CACTUS package, to summarize inferred mutations at each branch of the underlying Nymphalid phylogeny. We calculated rates for transposition ($d_P$), insertion ($d_I$), deletion ($d_D$), inversion ($d_V$), and duplication ($d_U$) events per million years (Ma) of evolution, based on the inferred divergence estimates from the dated phylogeny (see below). We subsequently calculated the ratio between $d_D/d_U$, in order to better understand genome changes over time. The evolution of genome size was also assessed using an ancestral state reconstruction method implemented in the R package PHYTOOLS[102], using the dated phylogeny (see below). The maximum likelihood ancestral character state at each node was inferred across the phylogeny and the function CONTMAP was used to plot these continuous character traits onto the phylogeny in PHYTOOLS. To explore evolutionary conservation at individual alignment sites, PHYLOP scores were computed. A neutral model was first generated as implemented in HALPHYLOPTRAIN.PY script (CACTUS package) using protein-coding exon coordinates of H. melpomene. Using that neutral

model, a PhyloP score measure (CONACC) was computed for each species in the phylogeny. A non-overlapped sliding window of 10 bp was adopted and data partitioned according to coding, intronic, 5'UTR and 3'UTR regions.

*HALtree:*(Dple:0.116311,(Bany:0.191535,((Jcoe:0.110411,Mcin:0.145408)Nymphalinae:0.0653829,(Smor:0.112769,((Pdid:0.0862526,(Dpha:0.0840608,(Ptel:0.0697011,Diul:0.0719228)Anc06:0.0146215)Anc04:0.0072575)OtherGenera:0.014715,((Djun:0.0477356,(Avfl:0.0260274,(Avcr:0.017931,Avpe:0.0601256)Anc07:0.00813407)Agraulis:0.0412044)DioneAgraulis:0.0386522,((((Eisa:0.0299466,(Evib:0.0492453,Elam:0.135873)Anc12:0.0242382)Anc08:0.0125611,(Etal:0.0348716,(Eali:0.156189,Elyb:0.0926062)Anc13:0.0188805)Anc09:0.0137634)Eueides:0.0353598,((((Hric:0.0619393,((Hdem:0.00639924,Hert:0.00593405)Anc24:0.0119802,((Hsar:0.0119348,Hleu:0.0506969)Anc30:0.0052799,(Hant:0.0488034,((Hcon:0.0131266,(Hhew:0.0120935,Hsap:0.0338259)Anc44:0.000815969)Anc41:0.00164225,Hele:0.020269)Anc36:0.00339148)Anc31:0.00410383)Anc25:0.00830886)Anc20:0.000866929)Anc15:0.00136963,(Hper:0.0360645,Hcha:0.00858555)Anc16:0.0155907)SaraSapho:0.00400497,((Htel:0.0177582,x(Hcly:0.0998115,Hhor:0.0439953)Anc21:0.00829279)Anc17:0.00782117,(Hhec:0.0248922,(Hher:0.0180353,((Herd:0.0175693,Hpet:0.0237403)Anc32:0.00245664,(Heet:0.0585169,(Hhim:0.0150149,Hlat:0.038479)Anc37:0.00274527)Anc33:0.00190247)Anc26:0.00517796)Anc22:0.00378605)Anc18:0.00213766)Erato:0.00130854)EratoSaraSapho:0.0160146,(Haoe:0.0587325,((Hheb:0.0460356,(Hhie:0.0535236,(Hdor:0.0130507,Hxan:0.00759445)Anc27:0.0108595)Anc23:0.00913015)Doris:0.0191776,((Hege:0.0167879,(Hbur:0.0147922,Hwal:0.0377348)Anc28:0.00775203)Wallacei:0.0275654,(Hnat:0.0169856,(((Heth:0.0707586,((Helv:0.0168755,Hpar:0.0137324)Anc45:0.00344984,(Hhel:0.0137256,Hatt:0.0206498)Anc46:0.00195758)Anc42:0.0012847)Anc38:0.00153125,(Hmel:0.0133198,((Htim:0.0135762,Hheu:0.0249147)Anc47:0.00384919,(Hpac:0.0415195,Hcyd:0.00800452)Anc48:0.00215685)Anc43:0.00221913)Anc39:0.0052026)Anc34:0.00113488,(Hbes:0.0199138,(Hism:0.023401,Hnum:0.0122538)Anc40:0.00149248)Anc35:0.00105549)Anc29:0.00240676)SilaniformMelpomene:0.0168094)WallaceiMelpomeneSilaniform:0.00298822)DorisWallaceiMelpomeneSilaniform:0.00308376)AoedesDorisWallaceiMelpomeneSilaniform:0.00603061)Heliconius:0.0298615)HeliconiusEueides:0.0308005)DioneAgraulisEueidesHeliconius:0.0161658)Heliconiini:0.0764233)Heliconiinae:0.0739338)HeliconiinaeNynmphalinae:0.0436992)BanyHeliconiinaeNymphalinae:0.116311)Nymphalidae;

## Gene prediction and transcriptome annotation

NCBI SRA archive was explored to look for available RNA-seq data. The best SRA archives were selected, downloaded (Supplementary Data 2) and added to the data generated in this study to re-annotate genes for available annotations (*H. erato v. 1* and *H. melpomene v.2.5*), and annotate genes for also for available assemblies. To do so the following pipelines was implemented (Supplementary Fig. 2): raw RNA-seq read data were filtered using TRIMMOMATIC v0.39[103] (ILLUMINACLIP:$ILLUMINACLIP: 2:30:10; SLIDINGWINDOW: 5:10; MINLEN: 100), and multiple approaches (predicted coding genes, ab initio and de novo) were implemented and combined in a pipeline with the aim of obtaining the best from each approach to overcome their own limitations.

The prediction of coding genes was implemented first. Quality filtered reads from multiple tissues (when available) were pooled then used as training data, and as input data for ab initio and de novo predictors. Reads were mapped to its reference genome (see Supplementary Data 1) using STAR (alignIntronMax = 500,000; alignSJoverhangMin = 10). The resulting sorted BAM files were used as training for the BRAKER v2.1.5 pipeline[104], which combines GENEMARK-ES SUITE v4.30[105] and AUGUSTUS v3.4.0[106], along with the masked genomes, generated with REPEATMASKER v4.1.1[107], using the Lepidoptera database, and the alignment of protein sequences from model species (*Drosophila melanogaster*, *Bombyx mori*, *Bicyclus anynana*, *Danaus plexippus*, *H. erato*, and

*H. melpomene*) using EXONERATE v2.46.2[97]. BRAKER is an automated pipeline that aims to predict genes using iterative training of AUGUSTUS. GENEMARK-predicted genes are filtered and provided for AUGUSTUS training, followed by AUGUSTUS prediction, integrating the RNA-Seq and protein alignment information, to generate the final gene predictions. To the regular BRAKER run we added a further UTR annotation step [braker.pl --addUTR=on], which uses the RNA-Seq coverage data via the tool GUSHR v1.1.0 to add UTRs to the augustus.hints.gtf file; it does not perform training of AUGUSTUS or gene prediction with AUGUSTUS and UTR parameters. Unfortunately, because the generated output (augustus.hints.utr.gff3) presents many syntax errors, we tried to mitigate them with a custom python script (BRAKER-GUSHRGFF3TOBED12v1.0.PY) available at https://github.com/francicco/-ComparativeGenomicsOfHeliconiini.

To generate the de novo transcriptome assemblies, quality filtered reads were assembled using TRINITY v2.10.0[108,109] separately for each tissue. The generated contigs were subsequently aligned to the genome using MINIMAP2. Coordinates for the aligned contigs were used to extract nt sequences, and TRANSDECODER v5.5.0 (http://transdecoder.github.io/) (minimum amino acid length >50), using homologs from UNIPROT database[110] and Lepidoptera proteome (see below) found with DELTABLAST v.2.7.1+[111]; and PFAM v33.1 domains[112] with HMMSCAN v3.3.2[113] (e < 1e − 10), was implemented to annotate coding regions.

To generate the ab initio transcriptomes, tissues-specific reads were realigned to the genome using STAR (same parameters as before) and BRAKER predicted splice sites, and assembled using both STRINGTIE v2.1.3B[114] and CUFFLINKS v2.2.1[115–117]. The assemblies derived from different tissues (when available) were subsequently merged using STRINGTIE and the BRAKER annotation as a guide [−merge -G braker.gff3]. Coding sequences were annotated as for the de novo approach. For both the de novo and the ab initio procedures, chimera transcripts (*i.e.*, erroneously merged transcripts) were scanned for and corrected using a custom python script (IDENDITYFUSEDTRANSCRIPTS.PY) available at https://github.com/francicco/-ComparativeGenomicsOfHeliconiini.

The whole procedure generated four annotations per species, a predicted one, a de novo, and two ab initio annotations. These were combined and used with STAR to map pooled reads again to their own assembly. The BAM file that was subsequently obtained was used as input for PORTCULLIS v1.1.2[118] [--threshold 0.5] to generate a splice-site DB. Finally, all these elements (transcript and spice-site annotations) were combined together using MIKADO v2.3.3[119] [--scoring insects.yaml -bt UNIPROTDB+Lepidoptera −mode split].

For species without their own transcriptome data (RNA-seq) the procedure to annotate their genomes was as follows. For each one, all mRNA transcripts from all of the annotated species so far were mapped to its genome using MINIMAP2 and the resulting BAM file was given to BRAKER [--addUTR=on] as input. The number of transcripts mapped was in fact enough to mimic a PacBio Iso-Seq scenario, were lesser but longer reads are used to generate a transcriptomic annotation. The resulting annotation was subsequently fixed with IDENDITYFUSEDTRANSCRIPTS.PY, followed by the TRANSDECODER procedure mainly to adjust start and stop sites.

The final step of the transcriptomic annotation (Supplementary Fig. 3) was performed using the COMPARATIVE ANNOTATION TOOLKIT (CAT)[120]. CAT is a comparative annotation pipeline that combines a variety of parameterizations of AUGUSTUS, including Comparative AUGUSTUS, with TRANSMAP projections through whole-genome Cactus alignments to produce an annotation set on every genome in that Cactus alignment. Leveraging high-quality gene sets, CAT is an attempt to synthesize all of the possible methods of genome annotation, relying on transcript projection, transcriptome and proteome alignments, simultaneous gene finding, and single-genome gene finding with full-length cDNA reads. CAT can therefore project annotations to other genomes, augmenting predictions, adding species specificity and detecting gene family expansion and contraction. CAT was run using --augustus

--rebuild-consensus, using *E. isabella* as a reference species. This choice was made because: *i)* its high quality, highly contiguous genome; *ii)* shows one of the species with the highest number of genes, and iii) as outgroup to identify gene expansion/contraction specifically for *Heliconius* species.

## Intron evolution analyses

Intron length evolution was studied looking at both the interaction with transposable elements (TEs), and looking at the rate of gains and losses across the phylogeny. Intronic regions were extracted from the longest transcript of each gene model using the annotations, as in Cicconardi et al. 2021[20], and their sequences were used to annotate TEs with REPEATMASKER. The correlation between genome size and median intron length was also computed with the PGLS function in R from the CAPER package[121]. The function implements GLS models accounting for phylogeny. We included a variance covariance array representing the phylogeny within the comparative dataset of dimension three (vcv = TRUE, vcv.dim = 3). The branch lengths were transformed optimizing between bounds using maximum likelihood (lambda = 'ML'). For each species introns were divided into short and long based on their median values. Their relative scaling coefficients and intercepts were subsequently analyzed with SMATR[122]. The intron turnover rate was subsequently estimated using MALIN (Mac OS X version)[123]. The intronic position of the ~3000 scOGs were used as input to identify conserved intron sites and infer their conservation status in ancestral nodes. The turnover rate (gain and loss) was estimated at each node with MALIN's built-in model maximum-likelihood optimization procedure (1000 bootstrap iterations). The model was also used to estimate the posterior probabilities of intron presence, gain and loss in extant and ancestral nodes.

## Functional annotation and orthologous-group dynamic evolution

The longest protein/isoform per gene was selected from each of the 63 Nymphalid butterflies' annotations. Because less contiguous and complete genomes could impair the orthology inference (i.e.: inflate copy number with fragmented transcripts), the best assemblies were selected by contiguity (i.e.: N50) and completeness (i.e.: BUSCO scores), and used to explore parameters in the orthology inference, as implemented in BROCCOLI v1.1[124]. BROCCOLI was designed to infer, with high precision, orthologous groups (OGs) using a mixed phylogeny-network approach. It performs ultra-fast phylogenetic analyses first and secondarily builds networks of orthologous relationships. Then, using a parameter-free machine learning algorithm (label propagation), it identifies OGs from the network. It also has the ability to detect chimeric proteins resulting from gene-fusion events and assigns these proteins to the corresponding OGs. Once parameters were optimized [-kmer_size 300 -e_value 1e-40 -sp_overlap 0.1 -min_nb_hits 20 -min_weight 0.3] (Romain Derelle, personal communication) a first iteration of the analysis was run using the whole dataset as input. Chimeric transcripts where then identified. For the *E. isabella* transcriptome the chimeras where manually curated by blasting single proteins to the BLAST server and then used to set up a custom automatic parser in python (BROCCOLICHIMERASPLITDATAGATHER.PY; available at https://github.com/francicco/-ComparativeGenomicsOfHeliconiini), before a second iteration was executed with the generated data. From this orthology table, pseudo-paralog loci from the highly fragmented genomes were identified and excluded if the sequence was shorter than the average value, with one and three standard deviations used as thresholds for scaffold and sub-chromosome level assembly, respectively.

From each OG a putative functional annotation was performed by identifying both the protein domain architecture and GO terms. For each sequence, HMMER v3.3.2 (HMMSCAN)[125] was first used to predict PFAM domains, then DAMA v2 (Domain Annotation by a Multi-objective

Approach)[126] was applied to identify architectures, combining scores of domain matches, previously observed multidomain co-occurrence, and domain overlapping. Annotation of GO terms was assigned with two strategies: homology- and predictive-based. For the homology-based strategy putative protein sequences were searched against *Drosophila melanogaster* protein databases (FLYBASE.ORG) using the DELTABLAST algorithm, assigning the resulting GO term to the locus. For the predictive-based strategy the CATH assignments for large sequence datasets[127–129] was implemented. Briefly, each input sequence was scanned against the library of CATH functional family (FUNFAMS v4.2.0) HMMs[129] using HMMER3 to assign FUNFAMS to regions on the query sequence as implemented in CATH-GENOMESCAN.PL. Then, the GO annotations for a matching FUNFAM were transferred to the query sequence with its confidence scores, calculated by considering the GO term frequency among the annotated sequences. Finally, a non-redundant set of GO annotations was retained, making up the GO annotations for the query protein sequence[129].

To characterize OG evolution, expansions and contractions were modeled as implemented in CAFE v5.0[27] based on OGs predicted with BROCCOLI. Because the model can be biased by incomplete genomes the analysis was performed using only genomes with complete BUSCO genes ≥90% (52/64 species). This allows us to account for assembly and annotation errors of the analyzed genomes. CAFE was run with the ultrametric species divergence tree (generated with MCMCTREE, see below), with 1-parameter model, estimating λ and α values and final run with estimated parameters ten times each to check convergence in order to produce robust parameter estimates and results (Supplementary Data 6).

## Selection on single-copy ortholog groups

To investigate the different evolutionary trajectories across Heliconiini, a genome-wise scan for genes present in at least *Eueides*, Erato1', 'Erato2', 'SaraSapho1', 'SaraSapho2', 'Aoede', 'Doris', 'Wallacei', 'Melpomene/Silvaniform' was implemented with a pipeline similar to that in Cicconardi et al.[3,130,131]. where the signature of diversifying positive selection was assessed by computing $\omega$ (the ratio of nonsynonymous to synonymous substitution rates; $d_N/d_S$) on five branches of the Nymphalid phylogeny (Heliconiini; *Dione + Agraulis + Eueides + Heliconius*; *Eueides + Heliconius*; *Eueides*; *Heliconius*), using codon-based alignments of groups of one-to-one (single-copy) orthologs (scOGs). More specifically, the nt sequences from each scOGs were extracted and a filtering procedure implemented before and after the alignment as follows. PREQUAL v1.02[132] [-pptype all] was used to filter nt sequences before the alignment, which was performed with MACSE v2.03[98] and filtered subsequently with HMMCLEANER[133] and GBLOCKS v0.91b[134] under a "relaxed" condition[135]. A ML gene tree was then generated as implemented in IQ-TREE2 v2.1.3 COVID-EDITION[136] [--sampling GENESITE -m MFP --keep-ident] and used in the adaptive branch-site random effects likelihood (ABSREL) method[137,138] as implemented in the HYPHY batch language[139].

Enrichment of GOTERMS was performed using a combination of two different approaches, the HYPERGTEST algorithm, implemented in the GOSTATS package[140] for R [annotation org.Dm.eg.db; conditional TRUE; testdirection over), and GOATOOLS[141] ($P$ value cutoff 0.05); both using scOGs with a putative sign of diversifying positive selection (adjusted $P$ value < 0.05), and as background all scOGs. To reduce false positive rate *conditional(p)* == TRUE (GOSTATS) was selected, a conditional algorithm that uses the structure of the GO graph to reduce subsequent tests[142], only considering terms in common between GOSTATS and GOATOOLS[141] results. GENEMANIA prediction server[143–145] was also used to predict functions of genes under selection. An interaction network methodology is implemented that, using information from protein and genetic interactions, pathways, co-expression, co-localization, and protein domain similarity, is able to determine enriched GOterms. An FDR cut-off of 0.05 was applied.

## Phylogenetic analysis & divergence estimates

Fully processed alignments of scOG were selected, concatenated and used to generate a maximum likelihood (ML) phylogenetic tree, as implemented in IQ-TREE2, partitioning the supermatrix for each locus and codon position, and with 5000 ultrafast bootstrap replicates, resampling partitions, and then sites within resampled partitions[146,147]; a strategy that may help to reduce false positives [--run 5 -B 5000 --sampling GENESITE -m MFP]. As a complement to the ML tree, gene trees from scOGs were retrieved from the selection analyses (see above), concatenated, and used to generate a coalescent summary method species tree, as implemented in ASTRAL-III v5.6.3[148]. As opposed to IQ-TREE2, ASTRAL-III can detect discordant topological signals in aggregated gene-trees, due to incomplete lineage sorting (ILS), and present a species tree topology accounting for that. ASTRAL-III completes incomplete input gene trees with respect to each other in order to define a bipartitions set as its search space[149], a different procedure was also adopted as implemented in TRIPVOTE v1.2[150], which works directly on a set of gene trees in an attempt to maintain a more faithful distribution of the gene trees discordance after completion. Therefore, ASTRAL-III was also run using this method. To further explore phylogenetic support, an analysis via quartet sampling (QS) was performed[151]. Briefly, QS provides three scores for internal nodes: (i) quartet concordance (QC), which gives an estimate of how sampled quartet topologies agree with the putative species tree; (ii) quartet differential (QD) which estimates frequency skewness of the discordant quartet topologies, and can be indicative of introgression, if a skewed frequency is observed, and (iii) quartet informativeness (QI), which quantifies how informative sampled quartets are by comparing likelihood scores of alternative quartet topologies. The QS analysis was run using the scOG supermatrix described above, specifying quartet likelihood calculations with 100 replicates (i.e., number of quartets draw per focal branch).

Finally, Bayesian algorithm of MCMCTREE v4.8A (from PAML package)[152] with approximate likelihood computation was used to estimate divergence times within the 64 Nymphalids. First, branch length by ML were estimated and then the gradient and Hessian matrix around these ML estimates were computed under MCMCTREE using the DNA supermatrix. As no fossils of Heliconiini or closely related tribes are known, we used four secondary calibration points (Supplementary Data 3; supplementary Fig. 21). These calibration points where selected based on their consistency with previous phylogenetic studies of Heliconiini and From the TIMETREE database[8,153]. For these priors a birth-death process with $\lambda = \mu = 1$ and $\rho = 0$ was used. In addition, a diffuse gamma-Dirichlet priors was given for the molecular rate ($\Gamma = 2{,}20$) and a diffusion rate ($\sigma2 = 2{,}2$). Ten independent runs were executed, each with a burn-in of 2,500,000 generations, sampling every 200, with nsample of 50,000. Convergence was checked using TRACER v1.7.1[154] making sure that the effective sample size (ESS) values were ≫ 200, and the uncertainties extracted from the posterior distributions of the Bayesian analysis

## Inferring introgression using gene tree-based method

To detect patterns of introgression within Heliconiini, we used two methods that rely on the topologies of gene trees for triplets of species. Given a species tree topology ((A,B),C), these tests are able to detect cases of introgression between A and C, or between B and C. These methods: discordant-count test (DCT), and the branch-length test (BLT), are implemented by Suvorov et al. (2021)[14] in a set of R scripts. They use complementary information—the counts of loci supporting either discordant topology, and the branch-length distributions of gene trees supporting these topologies, respectively—to test an introgression-free null model. In brief, DCT compares the number of genes supporting the two discordant gene trees: ((A, C), B) or (A, (B, C)); in the presence of (ILS) and/or in the presence of introgression. If the gene genealogies show either topology with equal

probability, then ILS is more expected to bias the topologies. In the presence of introgression, instead, one of the two topologies will show a higher frequency, mainly because the pair of species experiencing gene flow will be sister lineages. The BLT method instead examines branch lengths (substitutions per site) to estimate the most recent coalescence's age event. If under the concordant topology the coalescence is more recent than expected, then this should result from an introgression event. In contrast, with ILS, the coalescence should be older[155]. Furthermore, for a given triplet and for each gene tree the distance $d$ (a proxy for the divergence time between sister taxa) was calculated, averaging the external branch lengths leading to the two sister taxa under that gene tree topology[14]. These tests were applied on all triplets extracted from scOG gene trees within Heliconiini, and the resulting $P$-values were then corrected for multiple testing using the Benjamini–Hochberg procedure with a false discovery rate (FDR) cut-off of 0.05, using the provided R script BLT_DCT_FBRANCH_WRAPPER.R. We considered only introgression events that were significant in both tests. Because these tests are not independent, since different triplets may contain overlapping taxa, the correction results in more conservative tests[156], and the inferred FDRs may be somewhat inaccurate due to the statistical correlation of some triplets (A. Suvorov personal communication). Dsuite[156] was then used to plot the results in a heatmap plot[14] using the script GETTRIPLETSSUPPORTINGINTROGEVENTS.PY available at the GITHUB page of Dsuite.

## Landscape of local evolution history

To infer the distribution of phylogenetic trees across the whole clade of 63 butterfly species, we generated non-overlapping windows of 10 kb across the 63-way genome alignment, using the genomic coordinates of *E. isabella* as the reference. For each window HAL2MAF from the HALTOOLS SUITE v2.2[157] was used to convert the hal-format multi-genome alignment into Multiple Alignment Format (MAF), with the following options [--noAncestors --onlyOrthologs --noDupes] with conversion into a fasta format. Each block was filtered for gap regions using TRIMAL v1.4.REV22, retaining only alignment blocks with a minimum length of 2 kb and at least one non-*Heliconius* species and one member of each Heliconius clade. Each block was then used to generate the ML tree using IQ-Tree2, estimating extended model selection followed by tree inference with 1000 replicates for SH approximate likelihood ratio test. Finally, the resulting tree was then filtered out if the overall bootstrap values where <0.8. This was to ensure that each tree has a substantial robustness and phylogenetic signal. All valid trees from each ancestral chromosome and/or fused chromosome in *Heliconius* were then used to estimate the amount of introgression using the discordant-count test (DCT) and the branch-length test (BLT), as described above, and ILS as implemented in the coalescent based method, ASTRAL-III. For each method, we also compared trees from the Z chromosome versus all autosomal chromosomes combined.

## Evolution of gene families

Fifty-seven gene families spanning receptors, enzymes, channels, and transporters were selected to further explore the evolution of Heliconiini (Supplementary Data 4). From the functional annotation analyses of all the species, genes with a specific protein architecture were extracted and their amino acid sequences aligned using CLUSTALW v1.2.1[158]. The alignment was then used to build a ML tree for the whole gene family using FASTTREE [-lg -gamma -boot 1000 -fastest -pseudo] and used as input for MIPHY v1.1.2[159]. This tool can accurately predict members (orthologous group) of gene families, leveraging a species tree. Here, by visually inspecting all gene families it was used to annotate them to subsequently process OGs separately. For each gene family OG copy number was processed with CAFE (see above) to further explore events of expansion and contraction. Furthermore, each OG was analyzed to contrast differential evolutionary pressures between *Eueides* and *Heliconius* species. Each OG of all these gene families was

processed as follows: first, in the first step OGs subject to expansion (based on CAFE results) were excluded, alignments and gene trees were generated as in the signature of selection pipeline (see below). Second, from the tree, nodes with low bootstrap support (<0.90) and only the putative in-paralog[160] with the shortest branch from its ancestor were retained (custom python script REMOVEINPARALOGFROM-TREE.PY) available at https://github.com/francicco/-ComparativeGenomicsOfHeliconiini. This procedure was implemented to generate near single-copy ortholog groups (nscOGs) to exclude paralog biases in follow-up analyses. NscOGs were then realigned separately using nt sequences for *Eueides ssp.* and *Heliconius ssp.*, as in the signature of selection pipeline (see below). In each species group the mean $\omega$ was estimated (hyphy acd Universal $FASTA MG94CUSTOMCF3X4 Global) as implemented in HYPHY. For each nscOG we tested if the gene experienced a differential selective regime testing for relaxation/intensification the *Heliconius* branches compared with the *Eueides* ones with RELAX[161], as implemented in the HYPHY batch language[139] (*P*-value threshold 0.05).

## Annotating conserved non-exonic elements

To identify conserved non-exonic elements (CNEEs), we used the 63-way whole genome alignment and the PHAST v1.4 package[162,163]. Using the *E. isabella* coding gene annotation we first extracted 4-fold degenerate sites using HAL4DEXTRACT with the flag –conserved; and extracted the alignment with HAL2MAFMP.PY with --noAncestors --noDupes flags, and used PHYLOFIT to estimate the initial neutral model. We reduced the dataset to a subset of 10 species representing the whole Heliconiinae phylogeny (*S. mormoria, Dryadula phaetusa, Dione juno, E. isabella, H. erato, H. charitonia, H. sara, H. aoede, H. doris, H. melpomene*), which significantly reduced computational time, making this step feasible. To assess the convergence of the model we performed five independent runs of PHYLOFIT using --init-random, the strand-symmetric version of the generalized time reversible model (SSREV), to check likelihood stability. We then corrected the estimated neutral models for base composition statistics using MODFREQS, using the whole genome alignment including all the ancestral nodes. Once the neutral model was calculated, using the whole alignment, we estimated rho (the expected substitution rate of conserved elements relative to neutrality) for each ancestral chromosome using PHASTCONS; combining the conserved and non-conserved models with PHYLOBOOT, and finally using the averaged models to predict conserved elements in PHASTCONS, using default parameters. Finally, after estimating conserved elements, we merged elements within 5 bp of each other into single conserved element, and excluded regions shorter than 50 bp, or with less than 50 species, and gaps in consensus sequence less then 50%. We used PhyloAcc-GT[43], which computes the maximum *a posteriori* (MAP) **Z** matrix (matrix of latent states), and two Bayes factors to test for acceleration at the *Heliconius* stem. A Bayes Factor 1 (BF1), defined as the Bayes factor that compares a null model (no acceleration allowed on any branch) to the test branch model (acceleration allowed only on *Heliconius* branches); and a Bayes Factor 2 (BF2) defined as the Bayes factor comparing the test model to the full model (acceleration allowed on any branch). BF1 identifies elements accelerated in *Heliconius* irrespective of the pattern in the rest of the phylogeny, whereas BF2 identifies elements with acceleration specific to *Heliconius* species. *Heliconius*-accelerated elements therefore correspond to those with BF1 ≥ 10, and BF2 ≥ 1.

## GO enrichment among CNEEs

To test for gene ontology terms (GO) of functional elements enriched in *Heliconius*-accelerated CNEEs, we used two approaches, *i*) two permutation approaches to account for possible biases towards particular gene functions; *ii*) and a genomic fraction approach. In both cases, for each gene in the reference genome (*E. isabella*) we first assign a regulatory domain implementing the 5 + 1 kb strategy, as implemented in

GREAT[164]. We then assign to each coding locus a regulatory domain consisting of a *basal* domain that extends 5 kb upstream and 1 kb downstream from its transcription starting site, which do not overlap between different loci, plus a further *extension* up to the basal domain of the nearest upstream and downstream locus, up to 1 Mb.

For the permutations tests (*i*) we computed the expected probability (namely the mean of the distribution) of overlap between the regulatory domains of genes of a specific GO term and 10,000 random aCNEEs datasets, using BEDTOOLS SHUFFLE (-chrom -chromFirst -noOverlapping-chrom -chromFirst -noOverlapping) and randomly selecting the same amount of accelerate CNEEs among all CNEEs. A binomial test was then implemented to generate a *P*-value ($Pr_{binomial}$ = observed overlaps | number of aCNEEs, expected overlaps]). For the second test (*ii*) we implemented the same procedure as in GREAT[164], which computes the binomial test over the total fraction of genomic regions associated with a given GO term ($Pr_{binomial}$ = observed overlaps | number of aCNEEs, fraction]). Multiple test correction was done with the Benjamini–Hochberg FDR (FDR_bh) and Bonferroni as implemented in python library STATSMODELS.STATS.MULTITEST. With both approaches we test all biological processes where at least two aCNEEs were overlapping with a regulatory domain of the given GO term. We used REVIGO[165] to summarize results.

## Spatial and gene enrichment

We also looked for genomic regions enriched for aCNEEs, irrespective of CNEE-gene annotations. To do this, we generated non-overlapping 100 kb sliding windows and computed the probability of observing aCNEEs based on the binomial distribution over the 10,000 permutations of randomly selected accelerated CNEEs among all CNEEs, where the number of trials is the number of aCNEEs in the window and the probability of success is the average of the permutations ($Pr_{binomial}$ = observed aCNEEs per region | number of aCNEEs, expected aCNEEs per region]). In the same fashion we calculated the excess of aCNEEs per regulatory domain using the distribution over the 10,000 permutations ($Pr_{binomial}$ = observed aCNEEs per gene | number of aCNEEs, expected aCNEEs per gene]). The *P*-values of both tests where then corrected as before, using FDR_bh as implemented in python library STATSMODELS.STATS.MULTITEST. In the same fashion we checked enrichment in CNEEs over ATAC peaks obtained from Belleghem et al. (2023), computing the binomial distribution over the 10,000 permutations of all the reshuffled CNEEs (BEDTOOLS SHUFFLE) ($Pr_{binomial}$ = observed overlap | number of CNEEs, expected overlap with ATAC peaks]). The ATAC peaks from brain and imaginal disc of 5th instar caterpillars of *H. erato* were mapped over the *E. isabella* genome using HALLIFTOVER with the 63-way whole genome alignment, merging the resulting regions up to 5 bp apart to counteract gaps in the alignment, using BEDTOOLS MERGE.

## Cocoonases annotation & analysis

The protein sequence of *Heliconius* cocoonases was obtained from Smith et al. 2018[63], while the sequence from *Bombyx mori* was downloaded from NCBI. These sequences were used as queries in EXONERATE to find the loci in all the 63 assemblies. The results were used as a template to manually correct the final annotation of the loci in all species. All the amino acid sequences obtained were processed with TOPCONS webserver v2.0[166] to identify the peptide signal and CDSEARCH webserver[167] to verify the presence and the completeness of the conserved domain (pfam00089). All nucleotide sequences were therefore filtered using PREQUAL aligned using MACSE. Finally, IQ-TREE2 [--ALRT 5000 -B 5000] was adopted to generate a phylogenetic ML tree of all loci. For each of the four main clades selective forces were computed. More precisely, all sequences from each of the clade were filtered realigned separately and HYPHY we compute model test and the overall $\omega$ (hyphy acd Universal $FASTA MG94CUSTOMCF3X4 Global), while the number of sites under purifying and positive

selection using the single-likelihood ancestor counting (SLAC) method[168] as implemented in the HYPHY batch language[139] (P-value threshold 0.05). Sign of diversifying positive selection was detected by scanning all internal branches of the whole cocoonases phylogeny using adaptive branch-site random effects likelihood (ABSREL) method[137,138] as implemented in the HYPHY batch language[139], and correcting for multiple tests using a final P-value threshold of 0.05.

The structural analyses were conducted on 181 full length sequences of the 233 of the entire dataset (~220 aa in length). For each sequence the N-terminus region was clipped identifying the peptide signal coordinates was identified with SIGNALP v5.0b[169], and aligning the resulted sequences with the closest homolog protein for which a crystal structure is available (pdb: 4AG1), identified by HHPRED server[170]. The resulting final sequence was then used as input for ROSETTAFOLD v1.1.0[171] to predict three-dimensional structures, building five models per sequence using default parameters. A multiple alignment between all the resulting sequences was performed using the PDBALN function of the Bio3D packadge of R project[172]. Subsequently, the residue positions of multiple alignment containing more than 170 gaps were eliminated, resulting in a total of 227 residue positions. A graph theory based analysis was performed for each 3D model belonging to the final data set, as implemented in Ruiz-Serra et al. 2021[173], used to evaluate 3D protein structures in CASP14 2021. Briefly, the method adopts graph-based metrics in order to capture both the local features of the predicted distance maps (strength) as well as to characterize global patterns of the molecular interaction network. We represent any carbon alpha as a node of the network and each intra-molecular interaction as an edge. In particular, a link between two nodes (residues of the protein) is defined only if the euclidean distance between their c-alpha atoms is lower than 12 Angstrom. The analysis was performed for each of 5 models predicted for each sequence and then the corresponding averaged network was taken into account for the graph theory-based descriptions. To this aim, the strength descriptor was calculated for each node $i$ (residue $i$) of the network (protein), which is defined as the sum of the weights associated to the edges of all first neighbors (which are defined as the nodes connecting the node i); known as Residue Interaction Networks (RINs)[174]. In this way, this approach not only captures topological information due to the number of the first neighbors, which could be described, more simply, via degree parameter, but also is able to involve into analysis the distance value for each residue pair, which is directly linked to the energetic contribution. More specifically, in order to take into account, the spatial organization of the residue-residue non-bonded intra-molecular interactions, a weight needs to be applied at each residue-residue connection with the inverse of their c-alpha distance. Therefore, calculating the strength parameter for each node, both topology and an approximated kind of energetic contribution are compactly considered in the same descriptor. Thus, using "strength" parameter as a local descriptor for each node (residue) of the network (protein structure), we can associate a single numeric value to each residue. After performing a multiple alignment among all the sequences of the data set, we obtain an $N \times M$ output matrix, where $N$ is the total number of the sequences and $M$ the total number of all residue position (i.e., the position of the residue between all sequences is present at least once). In our case $N$ is equal to 181 and $M$ is equal to 227. Using a specific residue descriptor allows us to define a multiple alignment matrix, whose dimensions are again N × M, where instead of each amino acid in each position there will be the corresponding value of the descriptor based on graph theory (that is the "strength" in this cas), which is calculated considering any structural model. In this way, by mapping the information from sequence to structure, we can associate the values of the same matrix column (positions of the residues) in the same structural region of the proteins involved in the study (a structural alignment can easily show this association). Starting from this matrix, where for each element of the matrix the value of strength is

reported, we performed a Principal Component Analysis (PCA) with the aim to project each $M$-dimensional vector (i.e., the set of strength values associated to each protein of the data set) into essential space (i.e., the PCA space), which, in this case, is only composed by two components covering about the 39% of the total variance. Therefore, after the PCA approach, each protein structure is described only by two numbers which represent the projection of any strength vector (which is representative of the whole structure) along the two first principal components. This approach has a twofold advantage. On the one hand, it provides information on the spatial arrangement of proteins that can be easily visualized (given the considerable reduction of the number of the original variables). On the other hand, the method allows to evaluate the contribution (loading) of each original variable (the strength values associated with any residue position) for each of the two principal components. Is important to note that this methodology is completely independent from the phylogenetic signal as it does not take into account any information from the sequences.

To evaluate the consistency between the phylogenetic signal and the structural information of the four loci (groups), we adopted a completely unsupervised approach evaluating how groups are separated from each other in the PCA space. To this end, we calculated the four distributions considering both the first and second component, performing a Kolmogorov–Smirnov (K-S) test[175] as implemented in the R function KS.TEST, in order to validate the ability of the analysis to blindly distinguish the groups. To consider together the contribution of the first two components, we calculate the product between the two loadings, thus identifying the most crucial residues which are most responsible for separating the groups into essential space. To better interpret the results from a more structural and functional point of view, we map the key residues into best representative 3D structures, which are defined as the closest real sequence to the centroid of each group (calculated through the coordinates of the first two principal components) (Coc1A: Dryadula_Dpha.Dpha1503G76.2; Coc1B: Silvaniformis_Hpar.Scf0001930G23607b.6; Coc2: Erato_Hlat.Hel_chr17_2G18 418.1; Coc2b: Silvaniformis_Hhel.Scf00000159G13686.1; Coc3: Silvaniformis_Hnum_KU925753.1_Hmel_cocoonase5a.t4). Finally, to gain insights into the protein mobility 20 resolved x-ray structures of the human chymase were structurally aligned with PYMOL v1.20[176] to the representative Coc1A structure and the previously identified regions of the cocoonases were compared with the B-factor of the x-ray model, as this score reflect the fluctuation of atoms about their average positions, providing important information about protein dynamics[177].

## Reporting summary
Further information on research design is available in the Nature Portfolio Reporting Summary linked to this article.

## Data availability
The genomic data generated in this study have been deposited in the NCBI database under accession codes: PRJNA686707; PRJNA686708; PRJNA686710; PRJNA686711; PRJNA686712; PRJNA686713; PRJNA686714; PRJNA686715. Short-read data for *A. vanillae* (Peru; ERR5235460), *E. lampeto* (ERR5235459), *E. vibilia* (ERR5235454), *E. aliphera* (ERR5235452), *E. lybia* (ERR5235468), *E. tales* (SRR8883890), *H. telesiphe* (SRR8883900), *H. clysonymus* (SRR4032079), *H. hortense* (SRR4032054), *H. hecalesia* (SRR8883898), *H. erato petiverana* (SRR4032055), *H. erato etylus* (ERR5235453), *H. peruvianus* (ERR5235458), H. eratosignis (ERR5235467), *H. demeter* (SRR8883893), *H. ricini* (SRR4032011), *H. leucadia* (ERR5235456), *H. antiochus* (ERR5235455), *H. eleuchia* (SRR3102171), *H. congener* (SRR3102172), *H. hewitsoni* (SRR3102337), *H. sapho* (ERR266262), *H. hecuba flava* (ERR1143583), *H. hierax* (ERR1143585), *H. xanthocles* (ERR1143626), *H. egeria* (ERR5235461), *H. burneyi* (SRR8883892) *H. wallacei* (ERR1143625), *H. besckei* (SRR8883889), *H. ismenius* (ERR1143586), *H. numata* (SRR8883908), *H. ethilla* (ERR260305), *H. hecale*

(SRR8883896), *H. atthis* (ERR5235451), *H. pardalinus butleri* (SRR8883891), *H. elevatus* (SRR8883894), *H. pachinus* (ERS977714), *H. timareta* (SRR3102172), and *H. heurippa* (ERR3653294), were downloaded from NCBI.

## Code availability

Custom code used for these analyses is available on GitHub https://doi.org/10.5281/zenodo.8220878.

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

## Acknowledgements

This article would not be possible without the great support of the great *Heliconius* community. We are also grateful to the environmental ministries in Peru and Panama for permission to collect and export samples and the STRI community for assistance in the field. F.C. would like to thank Ronald Mori Pezo for his great help in collecting *H. aoede* and *Podotricha telesiphe*; Angel Corpuz for various informatics support including a great patience; Ian Fiddes, Mark Diekhans, Glenn Hichey and Marina Haukness for their great support for Cactus and CAT; Gregg Thomas and Tim Sackton for they help with PhyloACC-ST, preparation and analysis; Federica Cattonaro, Davide Scaglione and Simone Scalabrin from IGA (Udine, Italy) for their fruitful discussion on the best sequencing strategy to perfom. F.C. and S.H.M. are grateful to the High-Performance Computing team at the Advanced Computing Research Center, University of Bristol for support. We also thank the University of Puerto Rico Sequencing and Genomics Facility INBRE Grant P20 GM103475 from NIGMS, a component of the NIH, and the Bioinformatics Research Core of the INBRE. Its contents are solely the responsibility of the authors and do not necessarily represent the official view of NIGMS or NIH. This work was primarily supported by NERC IRF (NE/N014936/1) and ERC Starter Grant (758508) to S.H.M., which supported the work of F.C. Additional funding came from NSF EPSCoR RII Track-2 FEC (OIA 1736026) (R.P.), NSF IOS 1656389 (R.P.), and a Puerto Rico Science, Technology & Research Trust catalyzer award (2020-00142) (R.P.).

## Author contributions

S.H.M. and F.C. conceived and designed the study. F.C. and S.H.M. collected species in the field. F.C. performed all the genome assembly, filtering, annotation, alignment and other downstream bioinformatic analysis, while the structural analyses were performed by F.C., E.M. and D.d.M. A.M.V., S.M.V.B., P.R., J.H., E.E., C.D.J., W.O.M., R.P., and A.M. provided additional data. Visualization: F.C. Funding acquisition: S.H.M. Writing – original draft: F.C. and S.H.M. Writing – review & editing: F.C., S.H.M., A.M., D.D.M., E.C.P.d.C., A.M.V., S.M.V.B., A.A.R., J.H., C.D.J., W.O.M., R.P.

## Competing interests

The authors declare no competing interests.
