## [Peer review file · Nature Communications]

REVIEWER COMMENTS

Reviewer #1 (Remarks to the Author):

In this manuscript, Cicconardi and co-workers report the sequencing and generation of 9 contiguous genome assemblies and 29 reference-assembled genomes for Heliconiini butterflies. The team used this new data together with existing genomes to calculate a new phylogenetic hypothesis and to explore genome size, mobile elements, and some candidate gene families.

How the new genomes were sequenced and annotated is sound and according to current standards in the field, making this work a useful resource for the Heliconius community. The phylogenetic analyses, on the other hand, are somewhat shallow and it seems that only the most necessary was made (and reported). Here, the authors missed a chance to go deeper into the matter and to provide robust hypotheses and explore uncertainties. The biological aspects of this study are mostly superficial, and I have difficulties to see the “new biological insights” that this study pretends to provide for traits such as mushroom bodies or enzymatic processes.

Conceptually, the authors place their work in the context of identifying the genetic basis of key innovations related to the adaptive radiation of Heliconiini butterflies (lines 57-63). I feel that this is a too big theme for what they really have to offer. Besides, I do not fully understand the basic assumptions here. Is the idea that these key innovations evolved in the common ancestor of Heliconius and enabled them to “radiate”? If so, then the “genetic basis” (and molecular signatures of selection) of such key innovations should be searched for at this specific branch.

The reconstruction of cocoonase evolution is an interesting case study that should serve as starting point for future functional analyses. The figures are generally of high quality. The methods section is in parts too brief (important information on e.g. the phylogenetic analyses [alignment lengths, number of markers, model parameters] or genome size determination are lacking).

Further comments:

59-60: What is the difference between “molecular” and “genetic” basis here?

98-101: Couldn't it be that the “basal Heliconiini” nodes are too ancient for gene flow to be detected with the two strategies applied?

113-115: How was genome size determined (I could not find this information in the manuscript)? From the sequencing data? If yes, how close are these estimated to real measurements? Is there not a risk of circularity in argumentation?

131-134: Should this correlation not be corrected for phylogeny?

372: How were uncertainties in divergence times treated here? Where these sampled from the probability distributions?

432-444: Please provide more information on e.g., alignment length, number of SNPs used for the phylogenetic analyses, search parameters, etc. Is there a reason why no BEAST analysis was performed?

Fig. 6: “c” is missing at the PCA plot.

Reviewer #3 (Remarks to the Author):

NCOMMS-22-43328 Review

I have now had to time to read through and reflect upon the MS by Cicconardi and colleagues. They present the results of a large-scale comparative genomics analysis in an iconic clade of butterflies, well known for several aspects of their natural history and functional genomic studies. Certainly, the data involved and the methods employed are of high quality, and there is a lot of work invested here. Importantly, there is the potential for advancing the field using this data and many of the results. However, at the moment I feel the paper lacks focus and it could employ additional methods to investigate adaptive dynamics associated with their focal key innovations. As focused now, it comes across a bit as new comparative genomics resource coupled with a fishing expedition for genes involved in key innovations. Thankfully, I think the paper could be significantly improved without too much effort by the authors and be a major contribution to the literature. Specifically, this paper would have a much higher impact if the authors more formally tested the hypothesis of key innovations arising from any genomic signatures, being more conclusive as to what is a potential sign, and what is not, of genes likely being involved with such phenotypes. Also, integrating some measures of CRE/regulatory region evolutionary change in their assessments of key innovations would be a substantial addition that I think the authors could easily add, at least for some of their focal genes and potentially genome wide analyses. Below I explore these issues in more detail, followed by minor comments by line number.

In sum, with some refocusing and additional analyses, this paper could make a substantial contribution to the literature and greatly impress the readership of Nature Communications.

Major concerns:

Focus of evolutionary analyses in relation to key innovation traits.

I agree with the authors, line: 59-60, that "Identifying and understanding the molecular and genetic basis of such key innovations is now a realistic goal". In this pursuit using their data, I would like to see the authors more clearly testing whether they can detect such signature. If the goal is to generate candidate loci as causal hypotheses, they must with circumspection assess whether there are, or are not, finding likely candidates via their panoply of approaches. Without such guard rails, efforts like the current MS come across as much hand waving, fishing expeditions as regardless of what is found by endless analyses in search of p-values, results are almost always interpreted in the most adaptive light. Such is Panglossian storytelling at its finest, and this should be discouraged. Global analyses of tempo and mode of genomic evolution are important (e.g. omega across gene categories, rates of gene expansions/contractions), and they avoid these problems, as they are genomic natural history reports. Getting the balance between them right is a challenge, especially when much of the global analyses are expected, and most of those would do better in SM than main figures.

One thing that strikes me is that upon reading through the SM, there is clearly a huge amount of thought and work behind all of the genomic integration. But that is really not conveyed to the reader. Can you find a way to make these efforts, and their importance, more clear to the reader? What is the added value of it, compared to other approaches? (e.g. does it allow for more insight into introgression dynamics? My hunch is no, so why are you giving space to introgression if there is nothing much new there? Instead find a way to leverage your novel contributions). Given all of the integration, can't you leverage it for more powerful analyses, such as regulatory region evolution? You give a sense of its power via comparative analysis of orthologous introns across the tree (scOG across the tree can be done without such integration), but if you were to chew upon the 5' and 3' regions having regulatory impacts, and indicate how these regions are changing across key nodes in your tree ... that's interesting (perhaps global evolutionary rate of these regions and if it shifts across node, or what genes it shifts for most). If you don't see a global shift across the Heliconius node, great, super interesting. Or perhaps 5' reg region or 3' reg region varies by GO categories, or is correlated with omega ... ah, that would be something to

look at ... omega vs. regulatory region rate change (or e.g. CONACC score for the various categories you assess in Fig.3). regardless of what you see, just the plotting of that reveals the real power of your efforts (again, scOG alignment across species is old hat (though filtering is not), so go further by comparing the output of such alignments with other comparative genomic measures that can only be obtained via your integrated alignment (e.g. CONACC score). For you latter identified GF of interest,, do they vary in CONACC score? Since global analyses of GF families that expand/contract might be very hard, can you look at this for specific clades of GF that appear important based upon other data (omega, expansion/contraction patterns, etc). Integrate your data for novel insights (and focused bioinformatic analyses).

Phylogenetic reconstructions.

I'm concerned about the use of a concatenated set of genes, comparative results using ASTRAL, and how these relate to genus level inferred species trees. I see no discussion of the relationship between the ML tree and the species tree from ASTRAL. My concern after reading is that for the youngest branches, where there is the most ILS and introgression dynamics, that is also where your results diverge from previous analyses. It is not made sufficiently clear how meaningful these discrepancies are, what the relationship is between the concatML vs. ASTRAL vs. previous results are, and why one set of results should be trusted over another. Not saying you need tons more in the main text, but something there (and then directing reader to SM if more details are needed). In sum, what's the added value here of your dataset over previous work? Just more confidence, or ...?

Temporal inferences.

It is not made sufficiently clear what your temporal estimates are actually based upon. Ideally, these would derive from fossils. Instead, what appears to be being used here are previous node estimates, studies standing on studies. Thus, there is little independent temporal inference being made here. Rather than forcing readers concerned about the robustness of temporal inference to go back through previous studies, the authors need to make clear what these calibration points actually are in the methods details of the SM. Are they derived from fossils, or Browers mtDNA 2%/M years from the early days, or just estimates from Hmel. mutation rates, followed by using those node estimates as if they are objectively meaningful? I'm sure the authors can easily clarify these issues and I ask that they do so.

L90-91: "indicating a burst", well, ... These results could also suggest that you have a poor fit to your temporal constraints, and if you do not sufficiently explore those, then this conclusion is overstated. After looking in your SM, SMF20 shows a nice posterior distribution across your tree, documenting that none of your nodes appear to be poorly fit. Great, there is evidence supporting your claim. Add SMF20 to this statement, perhaps along with, "given well fit posterior estimates per node (SMF2), these results are consistent with a burst" (ie. help the reader "trust" your results a bit more here).

L94. I'm used to seeing concordance (GFC, SFC) of gene tree datasets on the species tree inferred by ASTAL, but I don't see any such results. Rather all of your concordance measure seem to be from the Quartets analysis. Where are the ASTRAL results? How do the fit into things? I find SF19 to be confusing. I thought you ran ASTRAL on all the scOG genes, which should be many more than 63 sequences. This looks to be something quite different ... Regardless, this is a great figure structure for addressing the issue of methodological discordance in tree estimation. May I suggest this type of formal comparison of your MLconcat vs. ASTRAL on the genes trees from that concat dataset, to highlight for the reader where the conflict is in the tree, and provide the GCF scores on the ASTRAL tree (iQTree does this very well; <http://www.iqtree.org/doc/Concordance-Factor>). I'm sure these minor issue will be quick to address.

Introgression.

I'm a bit confused as to which trees are which across the paper. Ideally after Fig1, a single tree is considered the species tree, and this is used in the fbranch heatmap and the introgression analysis. Given now that the MLconcat tree is in Fig 1, and the astral is in fig2b, I'm not sure what tree is used in Fig2a (my god that is small). If you just said it was the same tree throughout, I'd not worry, but now it seems things are changing. Further, by using the ASTRAL tree here with coalescent units, temporal assessment of the introgression is very complicated for the reader. I don't see lots of temporal clustering of introgression, so you clarify why you are showing this? Moreover, what is the added benefit of using the ASTAL tree here and/or coalescent branch lengths? State it the text, or in this caption (could be that you are trying to shift over to the species tree from ASTRAL, and branch lengths there are not easy ... just use that tree as the fixed topology for your temporal analysis, easy fix). In sum, please revise to bring the reader along with you, to appreciate what you are trying to convey. If there is nothing much novel about what you have found, just be really honest about that, move things to the back and focus on the novel meat you have.

Also, I cannot find where the Pvalues are shown for all of the inferred DCT and BLT results, nor can I see where those two results are concordant in their identification of introgression (an important component of the original Drosoph paper, since the tests should compliment each other). Did you only use the intersection, or either, for your selection of sig loci? did you combine their pvalues before FDR? Also, not clear to me what the full set of gene trees used for those analyses actually were, and this matters (n=???; I've had variable results based upon number of trees before I filtered them for quality, by removing trees with too many 0 branch lengths, etc). Finally, it is not clear how you have taken the results from the DCT, BLT tests and plotted them using Dsuite. Please be more explicit about that integration. Finally for clarity, what was the threshold for drawing an arrow in Fig 2?

Finally, it is not clear at all how the inferred levels of introgression, their timing, and taxa involved are novel, concordant, or in conflict with the numerous previous studies on these dynamics. Great you can estimate these things with this many taxa. But if you are not going to spend sufficient space on presenting them in a meaningful way to the reader, perhaps cause there's just not much new to see here, I suggest you move this section entirely to SM, giving it one line saying, "We identified previous events, and few novel events, documenting that these new tree based approaches are suitable for larger scale comparative analyses (SMXXX)." And then move on to your novel findings. You could unpack things a bit, and look for concordance between using these two large dataset tests in comparison to more small sample size, traditional tests (e.g. PhyloNet). I've love to see how well they agree, but perhaps that's not the focus here. But it would get back to what should be your litmis test, that is, documenting how the scale of your dataset can function and give insights that are robust (hence this comparison between method insights) and novel (which is why you don't want to spend a lot of time here cause I don't think there's much novel)

Evolution of genome size.

I find the PhyloP scores of CDS, introns, etc. to be exactly as expected and not warranting including in the main text. If you had focused upon 5' or 3' regulatory region (e.g. 5kb), that could be interesting. Move Fig3b to SM (again, L124-128, if you are reporting an increase in more purifying selection, can't that be just increase in Ne ... not super interesting, unless you mention this more specifically. But still, SM material). Rather, the main novelty here appears to be the accordion dynamic of genome size. Thus, panel A needs to be greatly enlarged so people can see the tempo and magnitude of change. Currently Panel A branches are too thin for any meaningful interpretation.

L11-115 – unless you really lay it out for the reader, they are not going to appreciate what you've done here. More specifically, since in this paragraph you only talk about nodes with genome size change from 367 to 308, it reads as a big yawn for me. Instead, lead with the range of genome sizes in your dataset

(221 to 586)!!!!!! More than a doubling in size across your study species, and this is not due to genome duplication – that will floor non-butterfly people. Start with something like that, or how much variation you can find among sister species, or among species within a genus, and you'll have people's attention.

Fig3C. I'm confused. If branches are colored by rate, with blue being gains (green is better = GO), then the insert needs to be revised. Looks like Rate(del Mbps) is associated with blue. Just put "Rate ratio" and perhaps faster duplication, or faster deletion above each arrow, and then define "Rate ratio" in the figure caption. Make this inset verticle, and you'll have more space for genome size evolution panel.

Lines 141-142. Thank you for starting to compare your findings with the existing literature. More of this is greatly needed throughout the MS. But, *bombus* is not at a linnean clade level equivalent to *drosophilids* or *anophelines*. Perhaps talk about Hymenoptera and Diptera, so you are comparing insights at comparable levels general readers can appreciate.

CAFÉ analyses: Looks like you are running the newest version of CAFE, which I don't have experience with (has it been formally released?) so it's hard for me to know what sort of settings are and aren't default there, other than the root filter being the default in the new version. Is that the same root filter that was set in the studies you are comparing rates with (cause this filter has a huge effect!)? Would be nice to see proof of model convergence across runs since you haven't shown it. Are the rates vastly different between runs? Or only different by a little bit? Are the numbers you report averages across runs that have converged?

Fig. 4 and related analyses ... I honestly don't find this very interesting and I suggest moving it to SM. I agree you need to find ways to show the reader what you can do with your large, comparative dataset, but these results are just not it. Please see other comments on this issue.

Analyses of evolutionary dynamics / selection / candidate genes

These sections, from L:153 through to 267, I think could benefit from a refocusing. It is very difficult for this reader to follow the different clades, lists of various tests, different genes, in relation to different key innovations. The net effect comes across as a set of tests thrown at a list of phenotypes, with the best possible light used to interpret a highly variable set of results.

May I suggest a different approach? After a general description of selection dyanmcis across entire tree, and how this relates to ... e.g *Drosophila* comparative genomics, or other relevant clades where comparative genomic studies have been performed on large sets of taxa, focus upon your key innovations. This is what makes your study different, unique, etc.

Put your hypotheses forward, pull the gritty details back a bit, and unify how the results connect, ideally in a nice graphic. You have a glorious set of phenotypes for the key innovations of *Heliconius*. Great. Can your different tests come up with good candidate loci for those phenotypes? well, kinda, right? But with all your different clade level analyses, you're losing me. Keep it simple. 1) you have specific nodes of interest, which allow you to ask how traits and molecular signatures might change across them. That tells you about module building over time, or quick jumps in novelty. You have specific sets of tests, from codon based, to gene expansion/contraction, and hopefully regulatory region rate change. Great. Now integrate and be objective. For example, either there were expansions in genes happening solely at the key nodes, or not, or were already happening before the *Heliconius* evolved. Great. That's three separate outcomes fitting with how we understand key innovations to evolve, in that they are either built along the way, appear rather spontaneously, or don't involve gene birth/death, or they don't evolve via coding changes at all, but are rather regulatory based upon the genes that are already existing. I suggest

structuring your results that way. It helps readers know what tests you are using and why, and how your different nodes are important, and how those come together to reveal something about the various key innovations Heliconius is known for. In sum, explore the evolution of key innovations by trying to detect these types of signatures being associated with them. Walk through each phenotype and the evidence you have or don't for it (gene exp/loss, omega), whether these comparative genomic signatures of adaptive evolution are diffuse (across several nodes) or unique to Heliconius node. Fig 5 just does not work for me at all.

Stated another way, without a clear hypothesis testing framework and the rigor it provides for structuring tests and nodes, the current text is too cluttered. It also makes the hippo pathway work look very strange, while instead it is potentially an exceptional example of how the Heliconius brain is likely built as a Key Innovation, e.g. it was not something that just appeared at the Heliconus node, rather there were clear exaptations preceding (223-229). In contrast, cocoonase is a great example of a previous candidate from the literature that's not really well supported from the tests you are able run here. If you treat the cocoonase story as a hypothesis you are testing from the existing literature, it reveals the power of your approach to rule out candidates, or at least an axis of variation for those candidates (and I'd move Fig 6 to SM).

Conclusions

L271. It is not really clear what ambiguities you have resolved, since your work was not sufficiently compared to the existing literature across your various sections.

273-274. As I argue above, you have insufficiently focused on testing whether you have support for any of your key innovations. If you restructure to test each of your several Key Innovations more formally, then you can more emphatically state here that you have explore the extent to which various type of comparative genomic analyses support the different mechanisms by which diverse key innovations evolved.

L275-277. You are thinking along the lines of the stepwise construction. I'm suggesting you lean into that more and use it as an organizing principal to more formally structure your findings.

Summary:

I'm trying to encourage you to have more of the hypothesis testing framework, to organize your excellent and diverse work, and celebrate your ability to show the evidence for the step-side evolution of some traits, the absence of evidence for other traits (e.g cocoonase story), etc.

Finally, in an echo of what I stated in at the start, I think you are focusing too much upon the coding regions of genes in your search for adaptive evolution. Rather, what would be more powerful here is using your genomic alignments for detecting shifts in evolutionary rates of CREs mediating key network edges for pathways likely underlying your traits of interest, such as brain size. Hopefully this will be a bit of inspiration (Hu, Z., Sackton, T. B., Edwards, S. V., & Liu, J. S. (2019). Bayesian detection of convergent rate changes of conserved noncoding elements on phylogenetic trees. *Molecular Biology and Evolution*, 36(5), 1086–1100. <https://doi.org/10.1093/molbev/msz049>)

Minor:

L 4: "combined with improved ... " something seems to be missing in this sentence.

L97. Suggest starting new sentence at "we".

L435: please include in this statement that you were using the scOG gene trees for the ASTRAL analysis.

L536. Change “b” to “c”

REVIEWER COMMENTS

Reviewer #1 (Remarks to the Author):

In this manuscript, Cicconardi and co-workers report the sequencing and generation of 9 contiguous genome assemblies and 29 reference-assembled genomes for Heliconiini butterflies. The team used this new data together with existing genomes to calculate a new phylogenetic hypothesis and to explore genome size, mobile elements, and some candidate gene families.

How the new genomes were sequenced and annotated is sound and according to current standards in the field, making this work a useful resource for the Heliconius community. The phylogenetic analyses, on the other hand, are somewhat shallow and it seems that only the most necessary was made (and reported). Here, the authors missed a chance to go deeper into the matter and to provide robust hypotheses and explore uncertainties.

>> We thank the referee for pointing this out. Understandably it might seem we hadn't gone deeper into the phylogenetic analyses. Most of the results on the subject had been placed in the supplementary material to make the main text more concise. There was an extended section in the Supplementary Information where we described the analyses in more details, which we think represent the current state of the art phylogenetic methods for a dataset as large as the one we presented. We adopted a Maximum likelihood approach with gene tree/species tree reconciliation to generate a reliable species tree phylogeny and used quartet sampling (QS) to explore uncertainty across the phylogeny.

To address the reviewer's comment we have now moved more details into the main text, and better reference the additional supplementary data. In addition, we now also include new sets of analyses exploring the phylogenetic position of lineages that have particular importance for inferring the evolution of pollen-feeding and related trait, please see below.

The biological aspects of this study are mostly superficial, and I have difficulties to see the “new biological insights” that this study pretends to provide for traits such as mushroom bodies or enzymatic processes.

Conceptually, the authors place their work in the context of identifying the genetic basis of key innovations related to the adaptive radiation of Heliconiini butterflies (lines 57-63). I feel that this is a too big theme for what they really have to offer. Besides, I do not fully understand the basic assumptions here. Is the idea that these key innovations evolved in the common ancestor of Heliconius and enabled them to “radiate”? If so, then the “genetic basis” (and molecular signatures of selection) of such key innovations should be searched for at this specific branch. The reconstruction of cocoonase evolution is an interesting case study that should serve as starting point for future functional analyses.

>> We thank the referee for the constructive criticism. We fully agree that searching for key innovations must use branch-specific tests at the inferred ancestral node. Indeed, for this reason, and in line with the reviewer's comment, we focused our main analysis of selection across the genome on a small set of focal branches, rather than considering every branch in the tree. We have edited the text to make this clear:

“From a genetic perspective, one fundamental question in understanding how adaptive radiations emerge, is if a significant amount of change, and sources of variability, originate prior to the acceleration in diversification, and whether this variation facilitates the subsequent adaptive radiation. This would be consistent with phyletic gradualism at a genetic level.” (lines 63-66)

In addition, in our revision we present new sets of analyses where we specifically test one of the most interesting aspects of Heliconius evolution, namely the monophyly of Heliconius species + the non-pollen feeding species H. aoede. We did this by exploring the topological landscape across the 63-way whole genome alignment, exploring the impact of alternative topologies, and how introgression and incomplete lineage sorting impact the species tree. These new analyses are laid out in a new section entitled “The origin of major Heliconius lineages and pollen-feeding”. We think that this section provides an important foundation for the understanding of a true key innovation.

The figures are generally of high quality. The methods section is in parts too brief (important information on e.g. the phylogenetic analyses [alignment lengths, number of markers, model parameters] or genome size determination are lacking).

>> We have sought to confirm to the journal guidelines for Methods, which state that the methods section should not exceed 3,000 words. We have however further extended the supplementary information to provide more detail, from pages 15 to 17, and Supplementary figures 19 to 26.

Further comments:

59-60: What is the difference between “molecular” and “genetic” basis here?

>> We changed the text in “understanding the genetic basis” to generate no confusion.

98-101: Couldn't it be that the “basal Heliconiini” nodes are too ancient for gene flow to be detected with the two strategies applied?

>> Our understanding of the methods we employ is that the split of Heliconiini, dated between 20 and 30 Mya, is not so deep as to underestimate introgression events. As an example, the same strategy and methodology was applied to 155

Drosophila genomes (Suvorov et al 2022, Current Biology), and introgression events were found that were dated to within this range (20-30mya). This shows that in principle the methods should be able to find introgression in that phylogenetic framework. We now added the following text to the main text:

“Note, the putative lack of introgression at basal node of Heliconiini is unlikely to be simply explained by a lack power in the statistical methods used to detect introgression. The Heliconiini split is dated between 20 and 30 Mya, and the same methodology, applied to the *Drosophila* radiation¹³, has identified introgression events dated over 20 My, suggesting that in principle the methods applied should be able to find introgression in our phylogenetic framework.” (lines 121-125)

113-115: How was genome size determined (I could not find this information in the manuscript)? From the sequencing data? If yes, how close are these estimated to real measurements? Is there not a risk of circularity in argumentation?

>> We indeed used assembly size as a proxy for genome size. Given the very high BUSCO scores, the assembly sizes should be very close to the actual genome size, but it is nevertheless important to clarify this. We added the following sentence to the manuscript:

“As a measure of genome size, we adopted the assembly size. Although this approach has some limitations, the high BUSCO scores, and the lack of correlation between assembly size and contiguity ($R^2 = 0.002$; $p = 0.05$) indicate that the great majority of the assemblies are complete, most of the smaller assembly sizes are unlikely to be artifacts of incomplete assembly, and the quality control during assembly ensured that larger genomes were not due to DNA contamination. Therefore, assembly size should closely correlate with the actual genome size, and no circularity or biases should be present.” (lines 574-579)

131-134: Should this correlation not be corrected for phylogeny?

>> We thank the referee for the question. We now have a correlation correcting for phylogeny. The supplementary methods now state:

“Correlation between genome size and median intron lengths was also computed with the PGLS function in R from the CAPER package⁵⁷. The function implements GLS models accounting for phylogeny. We included a variance covariance array representing the phylogeny within the comparative dataset of dimension three ($v_{cv}=TRUE$, $v_{cv}.dim=3$). The branch lengths were transformed optimising between bounds using maximum likelihood ($lambda='ML'$).” (lines 321-325)

And added the following text to the caption of supplementary Figure 28:

“The correlation was also corrected for phylogenetic signal suggesting that intron length is associated with assembly size independent of phylogenetic effects (F-statistic = 24.65 on 1 and 61 DF; P-value = 5.853e-06; r² = 0.2761).”

372: How were uncertainties in divergence times treated here? Where these sampled from the probability distributions?

>> We thank the referee for the question. Yes, the uncertainties come from the posterior distributions of the Bayesian analysis. We clarify this in the supplementary methods with the following text:

“Convergence was checked using TRACER v1.7.1 making sure that the effective sample size (ESS) values were >> 200, and the uncertainties extracted from the posterior distributions of the Bayesian analysis”. (lines 427-429)

432-444: Please provide more information on e.g., alignment length, number of SNPs used for the phylogenetic analyses, search parameters, etc.

>> We thank the referee for this suggestion. We added more information to the results section of the phylogenetic analyses as follow:

“To generate the species tree, we first compiled a total data set of 4,011,390 base pairs of aligned DNA obtained from scOGs. The alignment has over 1.5 million parsimony-informative, ~500k singleton sites, and 1.9 constant sites.” (lines 95-97)

Is there a reason why no BEAST analysis was performed?

>> Yes, unfortunately the amount of data is not suited for BEAST. Given the high volume of data the BEAST mcmc algorithm would take an unsustainable amount of time to run.

Fig. 6: “c” is missing at the PCA plot.

>> DONE.

Reviewer #3 (Remarks to the Author):

NCOMMS-22-43328 Review

I have now had to time to read through and reflect upon the MS by Cicconardi and colleagues. They present the results of a large-scale comparative genomics analysis in an iconic clade of butterflies, well known for several aspects of their natural history and functional genomic studies. Certainly, the data involved and the methods employed are of high quality, and there is a lot of work invested here. Importantly, there is the potential for advancing the field using this data and many of the results.

>> We thank the referee for these positive comments and the appreciation of the work involved.

However, at the moment I feel the paper lacks focus and it could employ additional methods to investigate adaptive dynamics associated with their focal key innovations. As focused now, it comes across a bit as new comparative genomics resource coupled with a fishing expedition for genes involved in key innovations. Thankfully, I think the paper could be significantly improved without too much effort by the authors and be a major contribution to the literature. Specifically, this paper would have a much higher impact if the authors more formally tested the hypothesis of key innovations arising from any genomic signatures, being more conclusive as to what is a potential sign, and what is not, of genes likely being involved with such phenotypes. Also, integrating some measures of CRE/regulatory region evolutionary change in their assessments of key innovations would be a substantial addition that I think the authors could easily add, at least for some of their focal genes and potentially genome wide analyses. Below I explore these issues in more detail, followed by minor comments by line number.

In sum, with some refocusing and additional analyses, this paper could make a substantial contribution to the literature and greatly impress the readership of Nature Communications.

>> We thank the referee for the constructive suggestions. We have re-written a substantial part of the paper to emphasise the hypothesis-oriented analyses more clearly. We also employed additional methods such as new phylogenetic analyses (phylogenetic/introgression/ILS landscape) and implemented a whole new analysis on putative non-coding regulatory elements as suggested by the referee.

Major concerns:

Focus of evolutionary analyses in relation to key innovation traits.

I agree with the authors, line: 59-60, that "Identifying and understanding the molecular and genetic basis of such key innovations is now a realistic goal". In this pursuit using their data, I would like to see the authors more clearly testing whether they can detect such signature. If the goal is to generate candidate loci as causal hypotheses, they must with circumspection assess whether there are, or are not, finding likely candidates via their panoply of approaches. Without such guard rails, efforts like the current MS come across as much hand waving, fishing expeditions as regardless of what is found by endless analyses in search of p-values, results are almost always interpreted in the

most adaptive light. Such is Panglossian storytelling at its finest, and this should be discouraged. Global analyses of tempo and mode of genomic evolution are important (e.g. omega across gene categories, rates of gene expansions/contractions), and they avoid these problems, as they are genomic natural history reports. Getting the balance between them right is a challenge, especially when much of the global analyses are expected, and most of those would do better in SM than main figures.

>> We appreciate the reviewers' point here and have attempted to clarify our analyses in this regard. However, we would also argue that one benefit of data rich genomic resources is in hypothesis generation, as well as hypothesis testing. Our analyses reveal some very strong candidate genes with patterns of evolution consistent to what we expect of genes involved in some of the complex adaptations involved in this radiation. Rather than storytelling, these hypothesised gene-phenotype links lay the foundation for a wide range of future experimental work. Hence, we believe our analyses both test and create hypotheses, providing much to reflect and build on in the future.

One thing that strikes me is that upon reading through the SM, there is clearly a huge amount of thought and work behind all of the genomic integration. But that is really not conveyed to the reader. Can you find a way to make these efforts, and their importance, more clear to the reader?

>> We thank the referee for acknowledging how much effort was put into the manuscript. To address the referee's request, we hope the changes we made, as detailed below, will better convey to the reader the importance of the study.

What is the added value of it, compared to other approaches? (e.g. does it allow for more insight into introgression dynamics? My hunch is no, so why are you giving space to introgression if there is nothing much new there?)

>> We thank the referee for raising this point. On the one hand, we believe it is difficult to discuss the adaptive radiation of this tribe without making the phylogenetic picture clear, as introgression has played a significant role in the evolution of some trait diversity in *Heliconius* in particular, and the prevalence of these patterns also impact downstream analyses (e.g. inferring selection of branches where the gene trees have different topologies). However, we also argue that our findings are still relevant because usually these analyses have been done within clades and with smaller dataset (fewer genes, or fewer species). Our analysis is the first to date to include almost all the *Heliconius* species and their sister genera; and for the first time we evaluate possible evidence of introgression across phylogenetic scales. We made several changes, both in terms of style and content. There is now a new section "The origin of major *Heliconius* lineages and pollen-feeding" that not only explores multiple topologies at the base of the *Heliconius* radiation, but also evaluates introgression among different chromosomes, and the difference between autosomes and Z chromosome. We believe these analyses are particularly

beneficial for readers unfamiliar with the system, but that readers familiar with the Heliconius literature will also find these new analyses interesting, offering new contributions to the subject.

Instead find a way to leverage your novel contributions). Given all of the integration, can't you leverage it for more powerful analyses, such as regulatory region evolution?

>> We deeply appreciate these suggestions and did our best to incorporate reasonable analyses of putative regulatory regions in our revision. More details follow below.

You give a sense of its power via comparative analysis of orthologous introns across the tree (scOG across the tree can be done without such integration), but if you were to chew upon the 5' and 3' regions having regulatory impacts, and indicate how these regions are changing across key nodes in your tree ... that's interesting (perhaps global evolutionary rate of these regions and if it shifts across node, or what genes it shifts for most). If you don't see a global shift across the Heliconius node, great, super interesting. Or perhaps 5' reg region or 3' reg region varies by GO categories, or is correlated with omega ... ah, that would be something to look at ... omega vs. regulatory region rate change (or e.g. CONACC score for the various categories you assess in Fig.3). regardless of what you see, just the plotting of that reveals the real power of your efforts (again, scOG alignment across species is old hat (though filtering is not), so go further by comparing the output of such alignments with other comparative genomic measures that can only be obtained via your integrated alignment (e.g. CONACC score).

>> We thank the referee for these suggestions. Further analyses on UTRs are a very appealing, and we would hope to do these in a future study. However, it was unfeasible for the revision of this paper. Instead of adding new analyses on UTRs, we took another suggestion from the referee forward, which we believe is more robust and feasible given current constraints on time and resources. Therefore, we report the findings of new sets of analyses on conserved non-exonic elements/cis-regulatory elements in a new paragraph in the main text. (lines 311-389)

For you latter identified GF of interest, do they vary in CONACC score? Since global analyses of GF families that expand/contract might be very hard, can you look at this for specific clades of GF that appear important based upon other data (omega, expansion/contraction patterns, etc). Integrate your data for novel insights (and focused bioinformatic analyses).

>> We thank the referee for this comment. In fact, we did look at OGs within GFs but, instead of using CONACC scores, we adopted a more complex and statistically robust methodology (RELAX, Fig. 5b), to specifically test the relative selection/relaxation rate between Heliconius and its sister clade, Eueides. We tried to clarify this point by adding this text in the main text.

***“We further characterised OGs within each GF, aiming to test for correlated gene expansions/contractions and shifts in selective pressures between Eueides and Heliconius (see Supplementary Results for more details),[...]*” (lines 264-266)**

Phylogenetic reconstructions.

I'm concerned about the use of a concatenated set of genes, comparative results using ASTRAL, and how these relate to genus level inferred species trees. I see no discussion of the relationship between the ML tree and the species tree from ASTRAL.

>> We thank the referee for the comment. We now have improved this section by changing part of the text and included more supplementary figures. Please see the response to reviewer 1 above for more details.

My concern after reading is that for the youngest branches, where there is the most ILS and introgression dynamics, that is also where your results diverge from previous analyses. It is not made sufficiently clear how meaningful these discrepancies are, what the relationship is between the concatML vs. ASTRAL vs. previous results are, and why one set of results should be trusted over another. Not saying you need tons more in the main text, but something there (and then directing reader to SM if more details are needed). In sum, what's the added value here of your dataset over previous work? Just more confidence, or ...?

>> It is hard to make comparisons with other studies about ILS and topological consistency, mainly because previous studies always used a different species set. For instance, if one looks at the results of Edelman et al (2019, Science; “Genomic architecture and introgression shape a butterfly radiation”) they focus only on 15 Heliconius species and 2 other Heliconiini, and although their final topology, the one they believe is the “true” species tree for those species, is almost identical and does not clash with our reconstruction, the introgression events cannot be directly compared because our species set is more the 3 times richer. We believe that our contribution is more comprehensive and offers a broader view on the problem. Given the concerns of the referee, the phylogenetic section now contains more information, which we hope the reader will find beneficial.

Temporal inferences.

It is not made sufficiently clear what your temporal estimates are actually based upon. Ideally, these would derive from fossils. Instead, what appears to be being used here are previous node estimates, studies standing on studies. Thus, there is little independent temporal inference being made here. Rather than forcing readers concerned about the robustness of temporal inference to go back through previous studies, the authors need to make clear what these calibration points actually are in the methods details of the SM. Are they derived from fossils, or Browers mtDNA 2%/M years from the early days, or just estimates from Hmel. mutation rates, followed by

using those node estimates as if they are objectively meaningful? I'm sure the authors can easily clarify these issues and I ask that they do so.

>> We agree, this needed clarification. Our revised Supplementary Information now reads as follow:

“As no fossils of Heliconiini or closely related tribes are known, we used four secondary calibration points (supplementary table 3). These calibration points were selected based on their consistency with previous phylogenetic studies of Heliconiini, as documented in the TIMETREE database^{89,90}.” (lines 422-424)

L90-91: “indicating a burst”, well, ... These results could also suggest that you have a poor fit to your temporal constraints, and if you do not sufficiently explore those, then this conclusion is overstated. After looking in your SM, SMF20 shows a nice posterior distribution across your tree, documenting that none of your nodes appear to be poorly fit. Great, there is evidence supporting your claim. Add SMF20 to this statement, perhaps along with, “given well fit posterior estimates per node (SMF2), these results are consistent with a burst” (ie. help the reader “trust” your results a bit more here).

>> We thank the referee to help us clarify this aspect. It now reads:

“Interestingly, deeper branches of the phylogeny are characterized by high molecular substitution rates (Fig. 1c and Supplementary Table 3), indicating a series of bursts in evolutionary rate at the base of the radiation, supported by a highly sampled posterior distribution across our tree (ESS >> 1000; Supplementary Fig. 21).” (lines 105-107)

L94. I'm used to seeing concordance (GFC, SFC) of gene tree datasets on the species tree inferred by ASTRAL, but I don't see any such results. Rather all of your concordance measure seem to be from the Quartets analysis. Where are the ASTRAL results? How do they fit into things? I find SF19 to be confusing. I thought you ran ASTRAL on all the scOG genes, which should be many more than 63 sequences. This looks to be something quite different ... Regardless, this is a great figure structure for addressing the issue of methodological discordance in tree estimation. May I suggest this type of formal comparison of your MLconcat vs. ASTRAL on the genes trees from that concat dataset, to highlight for the reader where the conflict is in the tree, and provide the GCF scores on the ASTRAL tree (iQTree does this very well; <http://www.iqtree.org/doc/Concordance-Factor>). I'm sure these minor issue will be quick to address.

>> To clarify, as detailed in the methods we executed ASTRAL using all gene trees. This is presented in Fig. 2b. The branch lengths in that tree correspond to the gene concordance factor (GCF) / Coalescent units. In Fig. S19 we plotted the comparison between the concatenated ML analysis (left) and the concatenated ML using the Astral topology (right). The caption of the figure now reads:

“Supplementary Fig. 19 | Comparison of maximum likelihood (ML) concatenated species tree, and maximum likelihood (ML) concatenated dataset + (ASTRAL-III) species topologies. The number beside each node on the left tree indicates bootstrap support. On the right tree we used the concatenated dataset to infer branch length over the ASTRAL-III topology; bootstrap values plus the coalescent units (CU)/ gene concordance factor (GCF) shown to the right of the tree for Heliconius and deeper nodes”

We also now computed the gCF and sCF using Iqtree on both topologies and show it in a new figure S20.

Introgression.

I'm a bit confused as to which trees are which across the paper. Ideally after Fig1, a single tree is considered the species tree, and this is used in the fbranch heatmap and the introgression analysis. Given now that the MLconcat tree is in Fig 1, and the astral is in fig2b, I'm not sure what tree is used in Fig2a (my god that is small). If you just said it was the same tree throughout, I'd not worry, but now it seems things are changing.

>> Thank you for this observation. We always used the MLconcat tree unless it is specified otherwise. So for figure 2, all trees are MLconcat trees with the exception of Fig. 2b, as specified in the figure legend/text:

“b ASTRAL-III species tree derived from nucleotide gene trees, with mapped introgression events (red arrows) derived from the corresponding f-branch matrix. Dashed arrows indicate introgression events with lower support (triplet support ratio < 10%).”

Further, by using the ASTRAL tree here with coalescent units, temporal assessment of the introgression is very complicated for the reader. I don't see lots of temporal clustering of introgression, so can you clarify why you are showing this? Moreover, what is the added benefit of using the ASTAL tree here and/or coalescent branch lengths? State it the text, or in this caption (could be that you are trying to shift over to the species tree from ASTRAL, and branch lengths there are not easy ... just use that tree as the fixed topology for your temporal analysis, easy fix). In sum, please revise to bring the reader along with you, to appreciate what you are trying to convey.

>> The reason why we plot the introgression events on the ASTRAL tree (Fig. 2b) is because we wanted to show the relationship between the ILS and introgression on a temporal timeframe on a single diagram. The reader, could for example, see how the great majority of introgression events happen in clades that are relatively young. This information can also be more clearly deduced in Fig2c. The reader could also notice that there are more frequent events where coalescent units are low.

We tried to make it more clear to the reader changing the figure caption, which now reads as follow:

“Note how most of the introgression events not only happen within clades and among time overlapped nodes, but also how the majority of introgression events are affecting lineages with low CUs, indicating a lower barrier to gene flow.”

If there is nothing much novel about what you have found, just be really honest about that, move things to the back and focus on the novel meat you have.

>> We are well aware that given the amount of studies on introgression in Heliconius. Indeed, some of these introgression events we identify were known already, in which case it is reassuring that they are found using quite different analytical methods. But we also think that a complete picture over the entire tribe was missing. We detect some additional patterns of interest, for example the relative lack of introgression within the sara/sapho clade is completely new and unexpected, a result which is also strengthened by the strong difference in the distribution of coalescent units within the clade. Finally, including the non-pollen feeding lineage, H. aoede, in these introgression analyses is, we think, an important contribution to the overall story of the Heliconius radiation, as discussed in our response to reviewer 1 (above). Overall, we believe these analyses are informative about the Heliconiini radiation, and highly relevant for the community of scientists that are interested in introgression, and ecological barriers to it .

Also, I cannot find where the Pvalues are shown for all of the inferred DCT and BLT results, nor can I see where those two results are concordant in their identification of introgression (an important component of the original Drosoph paper, since the tests should compliment each other). Did you only use the intersection, or either, for your selection of sig loci? did you combine their pvalues before FDR? Also, not clear to me what the full set of gene trees used for those analyses actually were, and this matters (n=???; I've had variable results based upon number of trees before I filtered them for quality, by removing trees with too many 0 branch lengths, etc). Finally, it is not clear how you have taken the results from the DCT, BLT tests and plotted them using Dsuite. Please be more explicit about that integration. Finally for clarity, what was the threshold for drawing an arrow in Fig 2?

>> We now added a table with P-values to the SI section of the revision. Table S4 was generated with blt_dct_fbranch_wrapper.r, which is available through github. This method calculates the FDR for the two tests separately, and we only considered introgression events if both of these FDR values were below 0.05. The script getTripletsSupportingIntroEvents.py was used to generate the heatmap. As stated in the caption of the figure, the low supported arrows have triplet support < 10% in Table S5. We have now added this to the methods.

Finally, it is not clear at all how the inferred levels of introgression, their timing, and taxa involved are novel, concordant, or in conflict with the numerous previous studies on these dynamics. Great you can estimate these things with this many taxa. But if you are

not going to spend sufficient space on presenting them in a meaningful way to the reader, perhaps cause there's just not much new to see here, I suggest you move this section entirely to SM, giving it one line saying, "We identified previous events, and few novel events, documenting that these new tree based approaches are suitable for larger scale comparative analyses (SMXXX)." And then move on to your novel findings.

>> We respectfully disagree with the reviewer. For the reasons detailed above, we think this section is an important part of the data analysis, and helps establish the phylogenetic framework for subsequent analyses. We feel the space dedicated to it in the manuscript is appropriate for our purposes and interests.

You could unpack things a bit, and look for concordance between using these two large dataset tests in comparison to more small sample size, traditional tests (e.g. PhyloNet). I'd love to see how well they agree, but perhaps that's not the focus here. But it would get back to what should be your litmus test, that is, documenting how the scale of your dataset can function and give insights that are robust (hence this comparison between method insights) and novel (which is why you don't want to spend a lot of time here cause I don't think there's much novel)

>> We thank the referee for asking this. We actually spent quite a bit of time trying to implement other methods, but most tools are not created to handle this amount of data and either did not run or would have required unreasonable amounts of computational time to do so. We also tried PhyloNet, but it was unable to handle the size of the data. Other tools required all taxa to be present in the gene tree/alignment, but in our data sometimes some taxa were missing, a problem which is also more prevalent with larger datasets. We think the two approaches that we used worked perfectly for the type and amount of data we had and have additionally never been used to infer introgressions in *Heliconius*.

Evolution of genome size.

I find the PhyloP scores of CDS, introns, etc. to be exactly as expected and not warranting including in the main text. If you had focused upon 5' or 3' regulatory region (e.g. 5kb), that could be interesting. Move Fig3b to SM (again, L124-128, if you are reporting an increase in purifying selection, can't that be just increase in N_e ... not super interesting, unless you mention this more specifically. But still, SM material).

>> We thank the referee for suggesting this change, but the plot in Fig3b does not just show the CONACC scores for different genomic compartments, but importantly also shows the difference for each compartment (CDS, intron, UTRs) between *Heliconius* and *Eueides*. This reveals, at the macroscopic level, an interesting result, a significant shift in acceleration/conservation as detailed in lines XXX:

"Between the two genera, we identified an enrichment for higher CONACC scores in *Heliconius* for CDS and introns, compared to the same compartments in *Eueides*, a trend that is inverted for the two UTR regions (Wilcoxon rank-sum test

‘two-sides’ P value < 2.2x10⁻¹⁶). This suggests an increased tendency for clade-specific selection, also confirmed by the fast-unconstrained Bayesian approximation method (FUBAR), which showed that Heliconius have more sites under purifying selection and positive selection per codon compared with Euiedes (Supplementary Fig. 27).” (lines 196-201)

Rather, the main novelty here appears to be the accordion dynamic of genome size. Thus, panel A needs to be greatly enlarged so people can see the tempo and magnitude of change. Currently Panel A branches are too thin for any meaningful interpretation.

L11-115 – unless you really lay it out for the reader, they are not going to appreciate what you’ve done here. More specifically, since in this paragraph you only talk about nodes with genome size change from 367 to 308, it reads as a big yawn for me. Instead, lead with the range of genome sizes in your dataset (221 to 586)!!!!!! More than a doubling in size across your study species, and this is not due to genome duplication – that will floor non-butterfly people. Start with something like that, or how much variation you can find among sister species, or among species within a genus, and you’ll have people’s attention.

>> We thank the referee for the advice. We have now changed the text as follow:

“By reconstructing ancestral genome sizes at each node in the Heliconiini phylogeny, we found that the MRCA of Heliconiinae experienced a 30% contraction from ~406Mb at stem of Heliconiinae to ~282Mb for Melpomene/Silvaniform clade; while, at the same time, other branches leading to Philaethria, Dryadula, Dryas and Podotracha, and the Erato, Doris and Wallacei clades within the genus Heliconius, had independent expansions.” (lines 188-191)

Fig3C. I’m confused. If branches are colored by rate, with blue being gains (green is better = GO), then the insert needs to be revised. Looks like Rate (del Mbps) is associated with blue. Just put “Rate ratio” and perhaps faster duplication, or faster deletion above each arrow, and then define “Rate ratio” in the figure caption. Make this inset verticle, and you’ll have more space for genome size evolution panel.

>> We thank the referee for the advice, have changed the size of the inserts according to the referee’s suggestions. We used blue instead of green to provide a color blind friendly palette. The reds are an increase rate of deletions over duplication, blues show an increase rate of duplication over deletions. It is a different type of information compared with the ancestral state reconstructions.

Lines 141-142. Thank you for starting to compare your findings with the existing literature. More of this is greatly needed throughout the MS. But, bombus is not at a linnean clade level equivalent to drosophilids or anophelines. Perhaps talk about Hymenoptera and Diptera, so you are comparing insights at comparable levels general readers can appreciate.

>> We thank the referee for the clarification, but the *Bombus* split, which we refer to in the manuscript, was dated ~34Mya (Sun et al 2020) similar to the split of our tribe Heliconiini (~25), where we have 90% of the species. So, although the Linnean level is not equivalent, the phylogenetic framework is similar and, we believe, more appropriate that order level comparisons. We would like to make more comparisons but unfortunately there are not so many studies similar in scale to our, especially for insects.

CAFÉ analyses: Looks like you are running the newest version of CAFE, which I don't have experience with (has it been formally released?) so it's hard for me to know what sort of settings are and aren't default there, other than the root filter being the default in the new version.

>> Yes, it was published on *Bioinformatics* in the 2020. Doi: 10.1093/bioinformatics/btaa1022

Is that the same root filter that was set in the studies you are comparing rates with (cause this filter has a huge effect!)?

>> Yes, the default settings are the same as in previous version. That includes the filter at the root.

Would be nice to see proof of model convergence across runs since you haven't shown it. Are the rates vastly different between runs? Or only different by a little bit? Are the numbers you report averages across runs that have converged?

>> We thank the referee for offering us the chance to clarify more the analysis. We checked convergence by running the model 10 times for both the parameter estimation and final run. For the parameter estimation the $-\ln L$ converged to 216801.63236761 with the minimum 216801.62388142. For the parameter estimation while from 179505.25769845 to 179505.25770217 for the final run. We further clarified this point we added a new table in the supplementary and the following text in the supplementary methods section as follow:

“CAFE was run with the ultrametric species divergence tree (generated with MCMCTREE, see below), with 1-parameter model, estimating λ and α values and final run with estimated parameters ten times each to check convergence in order to produce robust parameter estimates and results.” (lines 367-370)

Fig. 4 and related analyses ... I honestly don't find this very interesting and I suggest moving it to SM. I agree you need to find ways to show the reader what you can do with your large, comparative dataset, but these results are just not it. Please see other comments on this issue.

>> We thank the referee for the suggestion, but we respectfully disagree. We think the section offer an interesting view and is complementary to the genome

size dynamics. It shows that the change in genome size not only influence the gene content but also the gene architecture by shrinking or expanding intronic elements. Offering the opportunity to appreciate how important intronic CNEEs really are. Furthermore, not many genomic studies offer the opportunity to understand these dynamics at such detailed scale. We would like to keep the paragraph and the figure in the main.

Analyses of evolutionary dynamics / selection / candidate genes

These sections, from L:153 through to 267, I think could benefit from a refocusing. It is very difficult for this reader to follow the different clades, lists of various tests, different genes, in relation to different key innovations. The net effect comes across as a set of tests thrown at a list of phenotypes, with the best possible light used to interpret a highly variable set of results.

May I suggest a different approach? After a general description of selection dynamics across entire tree, and how this relates to ... e.g Drosophila comparative genomics, or other relevant clades where comparative genomic studies have been performed on large sets of taxa, focus upon your key innovations. This is what makes your study different, unique, etc. Put your hypotheses forward, pull the gritty details back a bit, and unify how the results connect, ideally in a nice graphic. You have a glorious set of phenotypes for the key innovations of Heliconius. Great. Can your different tests come up with good candidate loci for those phenotypes? well, kinda, right? But with all your different clade level analyses, you're losing me. Keep it simple. 1) you have specific nodes of interest, which allow you to ask how traits and molecular signatures might change across them. That tells you about module building over time, or quick jumps in novelty. You have specific sets of tests, from codon based, to gene expansion/contraction, and hopefully regulatory region rate change. Great. Now integrate and be objective. For example, either there were expansions in genes happening solely at the key nodes, or not, or were already happening before the Heliconius evolved. Great. That's three separate outcomes fitting with how we understand key innovations to evolve, in that they are either built along the way, appear rather spontaneously, or don't involve gene birth/death, or they don't evolve via coding changes at all, but are rather regulatory based upon the genes that are already existing. I suggest structuring your results that way. It helps readers know what tests you are using and why, and how your different nodes are important, and how those come together to reveal something about the various key innovations Heliconius is known for. In sum, explore the evolution of key innovations by trying to detect these types of signatures being associated with them. Walk through each phenotype and the evidence you have or don't for it (gene exp/loss, omega), whether these comparative genomic signatures of adaptive evolution are diffuse (across several nodes) or unique to Heliconius node. Fig 5 just does not work for me at all.

Stated another way, without a clear hypothesis testing framework and the rigor it provides for structuring tests and nodes, the current text is too cluttered. It also makes the hippo pathway work look very strange, while instead it is potentially an exceptional example of how the Heliconius brain is likely built as a Key Innovation, e.g. it was not

something that just appeared at the Heliconus node, rather there were clear exaptations preceding (223-229). In contrast, cocoonase is a great example of a previous candidate from the literature that's not really well supported from the tests you are able to run here. If you treat the cocoonase story as a hypothesis you are testing from the existing literature, it reveals the power of your approach to rule out candidates, or at least an axis of variation for those candidates (and I'd move Fig 6 to SM).

>> We thank the referee for the suggestions. We believe we have significantly changed the style of the manuscript giving a more hypothesis forward cut in different aspects:

- **We changed and reorganised part of the Introduction to better orient the reader.**
- **We now include more phylogenetic hypothesis tests to explore topological differences for key lineages relating to the evolution of pollen-feeding. This also includes the suggestions from reviewer #3 on concordance factors.**
- **We reorganised the "Expansion and Contraction of Gene Content" paragraph to clarify our more hypothesis driven approach; connecting the unsupervised and supervised sections better, and trying to lead the reader through the different phenotypes and results.**
- **We included an entirely new section on putative cis-regulatory regions (CNEEs), a connection with ATAC-seq data and how the acceleration of some of them are correlated with a number of phenotypes in Heliconius. This will set the foundation of future studies for the understanding the regulatory networks of genes that might be at the base of the evolution of key innovations in Heliconius species.**

In general, we believe we significantly improved both the style and the content at the scientific level of the manuscript.

Conclusions

L271. It is not really clear what ambiguities you have resolved, since your work was not sufficiently compared to the existing literature across your various sections.

>> We thank the referee. We have toned down the sentence. It now reads:

"our work offers a comprehensive view to the evolutionary history of an enigmatic tribe of butterflies, the evolution of their genomic architectures, and provides the most thorough analysis of potential molecular changes linked to the physiological and behavioural innovations of a diverse group of butterflies. These gene-phenotype hypothesis, alongside our comprehensive dataset, provide new opportunities to test and derive causative links between molecular and trait innovations." (lines 480-485)

273-274. As I argue above, you have insufficiently focused on testing whether you have support for any of your key innovations. If you restructure to test each of your several Key Innovations more formally, then you can more emphatically state here that you

have explore the extent to which various type of comparative genomic analyses support the different mechanisms by which diverse key innovations evolved.

>> DONE. We thank the referee. We think we have accomplished that now.

L275-277. You are thinking along the lines of the stepwise construction. I'm suggesting you lean into that more and use it as an organizing principal to more formally structure your findings.

>> DONE. We thank the referee for the suggestion. We now have reshaped the section with a better structure.

Summary:

I'm trying to encourage you to have more of the hypothesis testing framework, to organize your excellent and diverse work, and celebrate your ability to show the evidence for the step-side evolution of some traits, the absence of evidence for other traits (e.g cocoonase story), etc.

Finally, in an echo of what I stated in at the start, I think you are focusing to much upon the coding regions of genes in your search for adaptive evolution. Rather, what would be more powerful here is using your genomic alignments for detecting shifts in evolutionary rates of CREs mediating key network edges for pathways likely underlying your traits of interest, such as brain size. Hopefully this will be a bit of inspiration (Hu, Z., Sackton, T. B., Edwards, S. V., & Liu, J. S. (2019). Bayesian detection of convergent rate changes of conserved noncoding elements on phylogenetic trees. *Molecular Biology and Evolution*, 36(5), 1086–1100. <https://doi.org/10.1093/molbev/msz049>)

>> We want to thank the referee for this interesting suggestion. We have added a whole new set of analyses on non-coding elements as the referee suggested, applying a newer version of PhyloACC. This, we believe, improved substantially the diversity of the analyses in the current study, while also illustrating the use of phylogenetically dense datasets such as ours using cutting edge analyses which have never before been implemented in non-model insect organisms. In brief, with these analyses we show how the acceleration on putative regulatory elements might have influenced and shaped the evolution of some of the key innovations. For example, we detect signatures of acceleration in regulatory domains gene that might be responsible for long-term memory and nervous system at the stem of *Heliconius* species.

Minor:

L 4: “combined with improved ... “ something seems to be missing in this sentence.

L97. Suggest starting new sentence at “we”.

L435: please include in this statement that you were using the scOG gene trees for the ASTRAL analysis.

L536. Change “b” to “c”

>> *We thank the referee, we fixed all the minor issues.*

REVIEWERS' COMMENTS

Reviewer #1 (Remarks to the Author):

The revised version of this manuscript is clearly improved, the structure of the text makes it much more readable, and many questions raised in the first round of reviews (by both reviewers) have been addressed, in particular with respect to phylogenetic analyses, patterns of introgression, genome size evolution and 'outlier' genes/loci. Most of my initial concerns have been clarified, for example, on phylogenetic analyses and the more ancient signatures of gene flow (or better, the lack thereof). I still feel that the authors oversell some of their results, for example, when they talk about the genetic basis of key traits - after all, what they provide is a list of candidate genes. I would suggest toning this down, or as the other reviewer put it, be honest about it. Overall, I find their work a useful resource for the community that deserves publication.

Further comments:

61: I do not think "evolutionary niches" makes too much sense here, it should be "ecological niches". In the context of radiations, however, colonization mostly refers to novel environments such as islands with empty niches.

141: remove "intriguing" here

208: P-value (with "-"), as elsewhere in the manuscript

395: no ", " after examples

404: two different versions of "-"

422: Can you define what you mean with "keystone innovation"? Is that different from an "evolutionary key innovation" and if so how?

466-468: This is a rather strong notion given that what is presented here is a list of "candidate genes".

685: program name also in small caps?

Reviewer #3 (Remarks to the Author):

The revised manuscript by Cicconardi et al is simply stunning. They have taken the comments of both reviewers to heart, using what was helpful from among our concerns, discarding what was not, and revised substantially. Specifically, they have used their exceptional dataset to substantially move the non-model field of invertebrate research into the new era, from the study of introgression, genomic evolution, to targets of selection both in the coding as well as CRE regions. Additionally, their writing throughout is now very coherent, with a strong red thread connecting each step and analysis. This is going to be landmark paper and I am very excited to see it in press.

Regarding specific issues, or editorial comments .. I have none. This is a very thorough revision and I heartily applaud the authors.

REVIEWERS' COMMENTS

Reviewer #1 (Remarks to the Author):

The revised version of this manuscript is clearly improved, the structure of the text makes it much more readable, and many questions raised in the first round of reviews (by both reviewers) have been addressed, in particular with respect to phylogenetic analyses, patterns of introgression, genome size evolution and 'outlier' genes/loci. Most of my initial concerns have been clarified, for example, on phylogenetic analyses and the more ancient signatures of gene flow (or better, the lack thereof). I still feel that the authors oversell some of their results, for example, when they talk about the genetic basis of key traits - after all, what they provide is a list of candidate genes. I would suggest toning this down, or as the other reviewer put it, be honest about it. Overall, I find their work a useful resource for the community that deserves publication.

>> *We thank the referee for the comments, we addressed his last point at the end of this reply.*

Further comments:

61: I do not think "evolutionary niches" makes too much sense here, it should be "ecological niches". In the context of radiations, however, colonization mostly refers to novel environments such as islands with empty niches.

>> *Done.*

141: remove "intriguing" here

>> *Done.*

208: P-value (with "-"), as elsewhere in the manuscript

>> *Done.*

395: no ", " after examples

>> *Done.*

404: two different versions of "-"

>> *Done.*

422: Can you define what you mean with "keystone innovation"? Is that different from an "evolutionary key innovation" and if so how?

>> *We opted for evolutionary key innovation.*

466-468: This is a rather strong notion given that what is presented here is a list of "candidate genes".

>> *We thank the referee for pointing this out. We have now toned down the sentence, which now reads: "[...] our analyses ultimately allowed us to narrow down candidate genes that could be further tested to explore more in depth the molecular architecture of key innovations in this enigmatic group of butterflies."*

685: program name also in small caps?

>> *Done.*

Reviewer #3 (Remarks to the Author):

The revised manuscript by Cicconardi et al is simply stunning. They have taken the comments of both reviewers to heart, using what was helpful from among our concerns, discarding what was not, and revised substantially. Specifically, they have used their exceptional dataset to substantially move the non-model field of invertebrate research into the new era, from the study of introgression, genomic evolution, to targets of selection both in the coding as well as CRE regions. Additionally, their writing throughout is now very coherent, with a strong red thread connecting each step and analysis. This is going to be landmark paper and I am very excited to see it in press.

Regarding specific issues, or editorial comments... I have none. This is a very thorough revision and I heartily applaud the authors.

>> *We truly thank the referee for his very generous and kind words!*